# Integrative modelling of tumour DNA methylation quantifies the contribution of metabolism

Mahya Mehrmohamadi[1,2,3,4], Lucas K. Mentch[5], Andrew G. Clark[4,6] & Jason W. Locasale[1,2,3]

Altered DNA methylation is common in cancer and often considered an early event in tumorigenesis. However, the sources of heterogeneity of DNA methylation among tumours remain poorly defined. Here we capitalize on the availability of multi-platform data on thousands of human tumours to build integrative models of DNA methylation. We quantify the contribution of clinical and molecular factors in explaining intertumoral variability in DNA methylation. We show that the levels of a set of metabolic genes involved in the methionine cycle is predictive of several features of DNA methylation in tumours, including the methylation of cancer genes. Finally, we demonstrate that patients whose DNA methylation can be predicted from the methionine cycle exhibited improved survival over cases where this regulation is disrupted. This study represents a comprehensive analysis of the determinants of methylation and demonstrates the surprisingly large interaction between metabolism and DNA methylation variation. Together, our results quantify links between tumour metabolism and epigenetics and outline clinical implications.

[1] Duke Cancer Institute, Duke University School of Medicine, Durham, North Carolina 27710, USA. [2] Duke Molecular Physiology Institute, Duke University School of Medicine, Durham, North Carolina 27710, USA. [3] Department of Pharmacology and Cancer Biology, Duke University School of Medicine, Durham, North Carolina 27710, USA. [4] Department of Molecular Biology and Genetics, Field of Genetics, Genomics and Development, Cornell University, Ithaca, New York 14853, USA. [5] Department of Statistical Sciences, Cornell University, Ithaca, New York 14853, USA. [6] Department of Biological Statistics and Computational Biology, Cornell University, Ithaca, New York 14853, USA. Correspondence and requests for materials should be addressed to J.W.L. (email: jason.locasale@duke.edu).

DNA methylation is a major epigenetic mechanism that determines cellular outcome by regulating gene expression and chromatin organization[1] in a manner more dynamic than previously appreciated[2]. Altered DNA methylation is frequently observed in cancers compared with corresponding normal cells[3–5]. For example, global DNA hypomethylation[6] and tumour suppressor silencing by DNA hypermethylation are two of the most well-characterized cancer-associated alterations common across many human malignancies[7]. In addition to hypo- and hypermethylation, cancer cells exhibit increased variability in DNA methylation across large portions of the genome compared with their corresponding normal tissues[8,9]. Previous studies have shown that for several cancer types, variation in methylation levels among tumour samples is significantly higher than normal samples of the same tissue of origin[3,9], possibly indicating that deregulated epigenetics provides tumour cells with potential adaptive advantages[5]. While inter-tissue variability in DNA methylation is mainly explained by differentiation and tissue-specific regulatory mechanisms[10,11], very little is known about the functions and determinants of the high inter-individual variation among tumours of the same tissue type. Notably, a recent twin study on the determinants of inter-individual variability in DNA methylation reported that genetic difference among individuals account for only 20% of total variance with the remaining variance explained by environmental and stochastic factors that are yet to be identified[12].

The source of the methyl group for methylation is S-adenosylmethionine (SAM), which is generated from the methionine (met) cycle and is coupled to serine, glycine, one-carbon (SGOC) metabolism[13]. A large body of evidence indicates numerous roles for one-carbon metabolism in proliferation and survival of tumour cells through its roles in biosynthesis and redox metabolism[13–16]. The met cycle also mediates histone and DNA methylation in physiological conditions and provides a link between intermediary metabolism and epigenetics[17–19]. Although the network contributes methyl units to DNA, whether and to what extent this interaction is apparent in tumours and may contribute to cancer biology is unknown.

We set out to comprehensively quantify the contribution of various factors in explaining variation in DNA methylation. The advent of standardized genomics and other high-dimensional multi-platform 'omics' data through The Cancer Genome Atlas (TCGA) allows for systematic assessments of molecular features across cancers[20]. With combined statistical analysis, computational modelling and machine-learning approaches, we directly evaluated the quantitative contributions of molecular and clinical variables that lead to DNA methylation. We found a surprisingly large contribution for the expression of the methionine cycle and related SGOC network genes in explaining DNA methylation and identified numerous contexts, where this interaction may contribute to cancer pathology.

## Results

**Quantification of the determinants of DNA methylation.** It has been previously proposed that factors normally regulating the epigenome are disrupted in cancer, leading to increased variability of the cancer epigenome[5]. However, the nature and contributions of such factors are largely unknown. On analysis of global and local DNA methylation in tumours as measured by the Illumina Infinium HumanMethylation450K BeadChip arrays, we indeed found higher variation among tumours from the same tissue versus between different tissue types (Supplementary Note 1 and Supplementary Fig. 1a–d; Methods). Arrays were used over bisulfite sequencing because of the higher availability of these data in a standardized format allowing for an integrative analysis.

To establish quantitative relationships between DNA methylation and molecular and clinical features of tumours, we developed an integrative statistical modelling and machine-learning approach with the goal of identifying the relative contributions to within-cancer DNA methylation variation (Methods). We incorporated hundreds of variables into comprehensive statistical models of DNA methylation (Fig. 1a). Factors with a known role in DNA methylation machinery (chromatin remodelling enzymes and transcription factors), as well as factors with a potential biochemical link to DNA methylation (SAM metabolizing enzymes, met cycle enzymes and other SGOC enzymes that are connected to the met cycle[21]) were together considered (Fig. 1a). We also curated available clinical information such as age, gender and cancer stage in the calculations, where appropriate. Furthermore, since mutations are known to affect the cancer methylome[22], we included all recurrent genetic lesions (somatic mutations and copy number alterations) for each cancer type in our models. Together, over 200 variables were collectively analysed for each cancer type (Supplementary Data 1). Our models are therefore not completely agnostic as we pre-select classes of biological variables that are known to affect DNA methylation to avoid loss of statistical power by including too many features (for example, expression of all genes in the genome). Therefore, to test for potential bias, we also considered the expression levels of sets of random genes with functions non-related to DNA methylation as additional variables in our models (see Methods). Subsequently, we incorporated all variables into unbiased selection algorithms suitable for dealing with large numbers of prediction variables. For this task, we considered two independent approaches: a generalized linear model (Elastic Net[23]) and a machine-learning algorithm (Random Forest[24]). A distinct computation was carried out for each 10 kb genomic region with variable methylation (s.d. > 0.2) in each cancer type. Samples of each cancer type were divided into three independent test subsets and three training subsets, and separate models were generated using each subset. The models were then combined resulting in a single final model for each 10 kb region of DNA methylation in each cancer. Model performance was evaluated by measuring mean squared prediction error of test samples from Elastic Net and Random Forest separately (Methods).

We observed that our models predicted test set DNA methylation with small mean squared error (MSE < 0.04) in many regions across the genome (Fig. 1b). Comparison of the performances of the two methods showed that Random Forest and Elastic Net algorithms were able to predict DNA methylation with comparable MSEs on average (Fig. 1c and Supplementary Fig. 2a). In general, predictability of local DNA methylation was largely dependent on cancer type as well as chromatin region in each model. For example, we observed that local DNA methylation was most predictable in prostate and lung cancers and least predictable in liver and bladder cancers (Fig. 1c and Supplementary Fig. 2a). Together with the high variation in local DNA methylation levels seen in liver and bladder cancers (Supplementary Fig. 1d), these results suggest a higher stochasticity in the epigenetic signatures for these two cancer types compared with others in this study. On annotating genomic regions, where local DNA methylation could be predicted with a low error (MSE < 0.04) in each cancer, we found that the majority of the predictable regions lie within 20 kb of the transcriptional start site (TSS) of a gene (Supplementary Fig. 2b), suggesting that regulation of DNA methylation by the factors included in our models is stronger at genic regions.

We next performed a set of tests to evaluate the robustness of our modelling approach. To this end, we compared the original gene expression variables included in our models with a group

of variance-matched randomly selected genes from the genome (see Methods) in their ability to predict DNA methylation. In the presence of both groups of gene expression variables (original and random), both Elastic Net and Random Forest models selected our original variables significantly more frequently than random genes (higher rank corresponds to higher contribution; Mann–Whitney $P$ value = 0.0007 for Elastic Net and < 0.0001 for Random Forest; Fig. 1d; see Methods). When the same test was performed in the presence of five additional common gene families (receptor tyrosine kinases (RTK), receptor serine kinases (RSK), Toll-like receptors (TLR), MAPK signalling (MAPK) and WNT signaling (WNT)), all but RTKs ranked significantly lower (Mann–Whitney $P$ value < 0.0001) than the original gene expression variables that we initially included in our models based on biological functions (Supplementary Fig. 3; Methods). Together, these tests provide additional validation and confirm that the Elastic Net and Random Forest algorithms are suitable for quantitation of variable contributions in determining DNA

methylation. Given that our models are not completely agnostic, we do not rule out the possibility of existence of potentially highly contributing factors other than the hundreds of variables that we considered (for example, RTKs). As such, the results should be interpreted in the context of relative contributions among the variables included and the abilities of these variables in predicting DNA methylation.

**Metabolism is a major predictor of DNA methylation in cancer.** Using the results of the integrative modelling, we next quantified the relative contribution of different functional classes of variables in explaining DNA methylation variation within each cancer type. For this, we measured two independent metrics, one using the Random Forest variable importance scores, and the other using a binary score for whether or not a variable was selected by the Elastic Net models (non-zero coefficient). For each variable, an overall importance score was calculated by averaging its

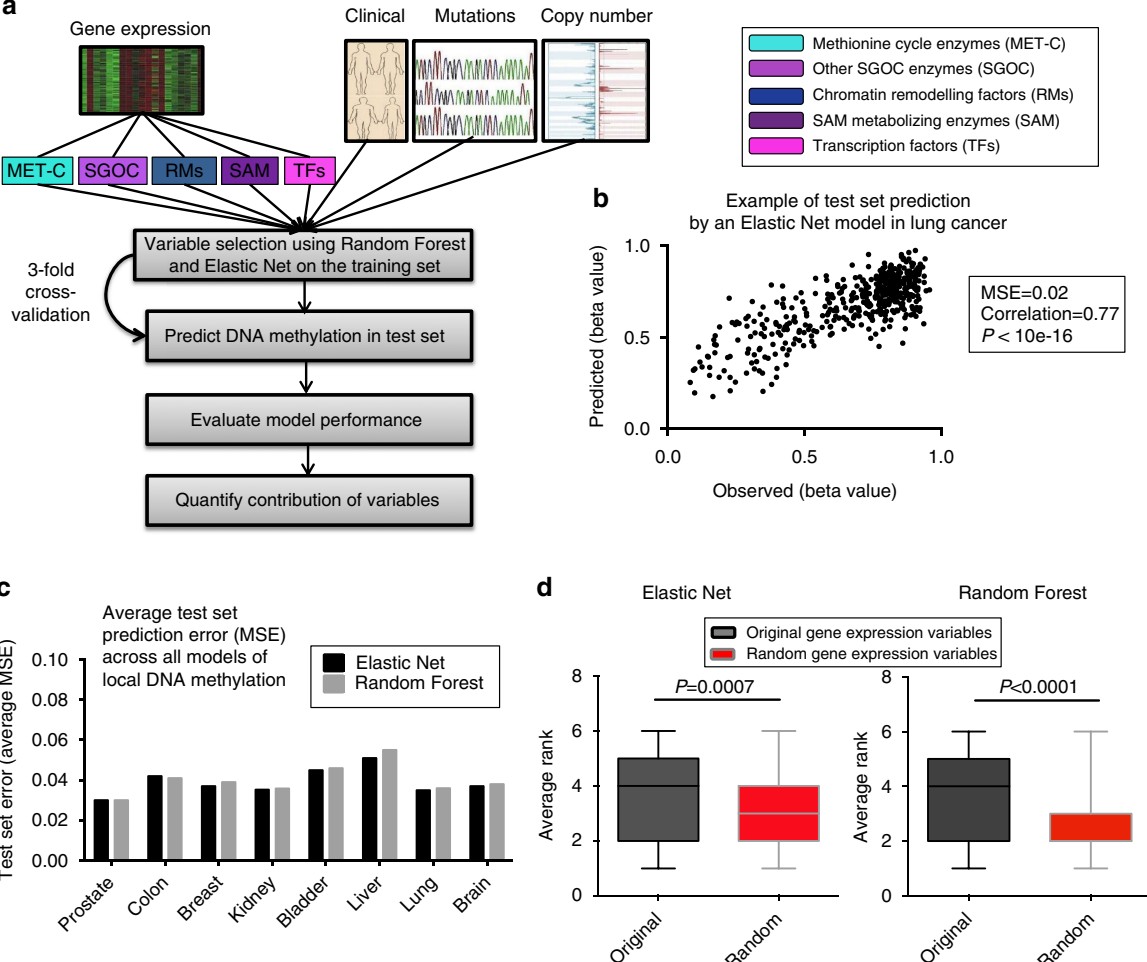

**Figure 1 | Integrative modelling of local DNA methylation levels.** (**a**) Schematic summarizing the integrative approach utilized for modelling local DNA methylations. DNA methylation at a given 10 kb region was predicted by incorporating relevant gene expression, somatic mutation, copy number alteration (GISTIC[56] values) and clinical information into integrative models (see Supplementary Data 1 for the complete list of variables included for each cancer type). (**b**) An example of an Elastic Net model performance in lung cancer. The x axis shows true values of DNA methylation in each sample, and the y axis shows the value predicted by the integrative modelling in the same sample when it was in the test subset. (**c**) Summary of overall model performance. For each cancer, the MSEs of test set predictions by Elastic Net and Random Forest were averaged across all models of local DNA methylation. (**d**) Comparison of original gene expression variables with randomly selected variance-matched genes. The y axis shows the average rank of each gene expression category based on average variable usage score across all Elastic Net models (left) and average variable importance score across all Random Forest models (right) of local DNA methylation in brain cancer (boxes extend from 25th to 75th percentiles, centre lines represent the median and whiskers show the minimum and maximum value in each group). P values associated with the Mann–Whitney test between the ranks across all models are shown (a higher rank corresponds to higher contribution; see Methods).

relative importance across all models of 10 kb DNA methylations, and an overall usage score was calculated by measuring the fraction of 10 kb regions in which Elastic Net models selected the variable (Methods). To estimate the contribution of each functional class of variables in explaining total variation in DNA methylation, we pooled all variables in the same functional category and averaged across their importance and usage scores separately (Supplementary Fig. 4a,b).

Results from both Random Forest and Elastic Net algorithms identified a considerable contribution from the variables within the SGOC metabolic network relative to other classes of variables ('other SGOC enzymes' was the second highest scoring among all classes, closely following 'transcription factors' according to both methods. 'methionine cycle enzymes' was the third and fourth according to Random Forest and Elastic Net, respectively) (Fig. 2a,b). Previous studies have shown that transcription factor abundance and occupancy strongly mediate dynamic DNA methylation turnover in regulatory regions[25,26]. Consistent with this observation, our results confirm the 'transcription factors' class has the highest contribution to predicting DNA methylation levels across human tumours. Notably, even in the presence of most if not all known variables that are thought to mediate the status of DNA methylation, metabolic factors still uniquely explained a large part of the variability in methylation (Fig. 2a,b).

Given the contribution of the methionine cycle and its biochemical link to DNA methylation, we further explored the variables within the met cycle class compared with all other variables in their ability to predict DNA methylation (Fig. 2c). Within the met cycle class, methionine adenosyltransferase 2 beta (MAT2B) and betaine-homocysteine S-methyltransferase 2 (BHMT2) exhibited higher predictive values than methionine synthase (MTR) and adenosylhomocysteinase (AHCY) on average (Supplementary Fig. 4c,d). Notably, in the presence of the nearly 200 other variables in the computations, the met cycle—especially MAT2B—still contributed substantially to DNA methylation prediction (MAT2B was ranked among the top 5% of highly selected variables in prostate, breast, liver, lung and brain cancers; Fig. 2d). We observed that the levels of MAT2B contribute to DNA methylation in nearly half of the variable regions across the genome even after accounting for various factors related to DNA methylation (MAT2B was selected by 42% of all Elastic Net models with MSE < 0.04 on average; Supplementary Fig. 4c). Together, our results confirm that metabolism contributes to DNA methylation in many cases of human cancer, and the association between metabolism and DNA methylation is stronger in some genomic regions than others.

**Functional annotation of metabolically regulated regions.** Results of the integrative modelling across cancers indicate that defined regulation of DNA methylation happens in regions where gene expression may be affected, thereby suggesting that this regulation could drive essential cancer biology. We next set out to

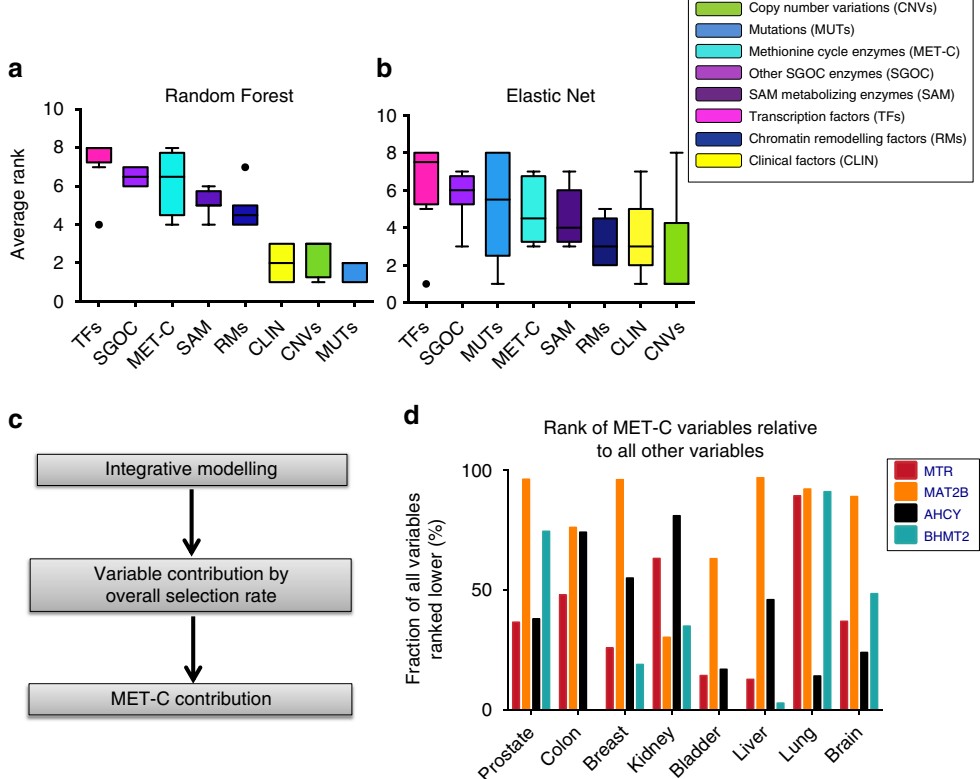

**Figure 2 | Contribution of different functional classes of variables to DNA methylation variation.** (**a**,**b**)Relative contributions of the variable classes according to Random Forest average variable importance (**a**) and Elastic Net average variable usage (**b**) are shown averaged across all cancers (Methods). The y axis shows the average rank of each class across cancers (with higher values corresponding to higher contribution). (Boxes extend from 25th to 75th percentiles, centre lines represent the median and whiskers show the minimum and maximum values in each class with the exception of individual outliers shown). (**c**) Diagram summarizing the steps taken towards calculating overall contribution of each of the met cycle variables relative to other variables in explaining variability in local DNA methylations. (**d**) Ranking all variables according to their overall selection rate (usage) across all models of local DNA methylation in each cancer. The y axis shows the per cent of variables that ranked lower than each of the met cycle variables (that is, made less contribution to DNA methylation) in each cancer (BHMT2 was removed from the models of colon and bladder cancers due to low expression).

characterize all regions across the genome where the association between DNA methylation and the met cycle activity is particularly strong. To identify such regions, we designed a scanning algorithm to locate genomic regions spanning multiple CpGs with significant peaks of correlation of methylation with expression of met cycle enzymes (Fig. 3a; Methods). We performed this analysis on each of the eight cancer types separately and identified distinct peak sets across the genome (Supplementary Data 2). To assess potential bias towards highly methylated regions and regions where there is higher probe density, we analysed the relationship between average absolute methylation of individual CpGs and their correlation with met cycle expression, and found no significant association ($P$ value of correlation $= 0.62$), confirming that the identified peaks are distinct from highly methylated regions (Fig. 3b; Methods).

Density plots of peak distributions relative to the TSS of the nearest gene were concentrated around the TSS in all cancers (Supplementary Fig. 5a), as expected given the higher density of probes in gene regulatory regions in the Illumina arrays (Supplementary Fig. 5b). However, by further visualizing the distribution of the peaks immediately surrounding the TSS, we observed that peak distributions are more diffuse around the TSS

(Supplementary Fig. 5c) compared with the probe density distribution control (Supplementary Fig. 5d). This suggests potential enrichment in areas of the genome overlapping with gene body regions and CpG island shores, where dysregulated DNA methylation has previously been observed in human cancers[5]. The peak distribution density plots extended up to a few hundred kilobases in distance from the nearest TSS, suggesting that DNA methylation at inter-genic parts of the genome may also be affected by the activity of met cycle.

We next tested the met cycle specificity of the identified peaks by correlating them with expression of randomly selected genes in the genome (Methods; Supplementary Fig. 6a). For the majority ($>83\%$) of the identified peaks, the met cycle's correlation with DNA methylation was significantly non-random ($P$ value $<0.05$; Supplementary Fig. 6b). These results show that our approach was able to identify genomic regions where DNA methylation levels are specifically affected by the met cycle activity.

We next set out to identify genes that overlap with the identified peaks in each cancer type. Functional annotation of genes overlapping these peaks by means of pathway enrichment analyses across a comprehensive collection of more than 70 gene-set libraries[27] showed enrichment of epigenetic features in these

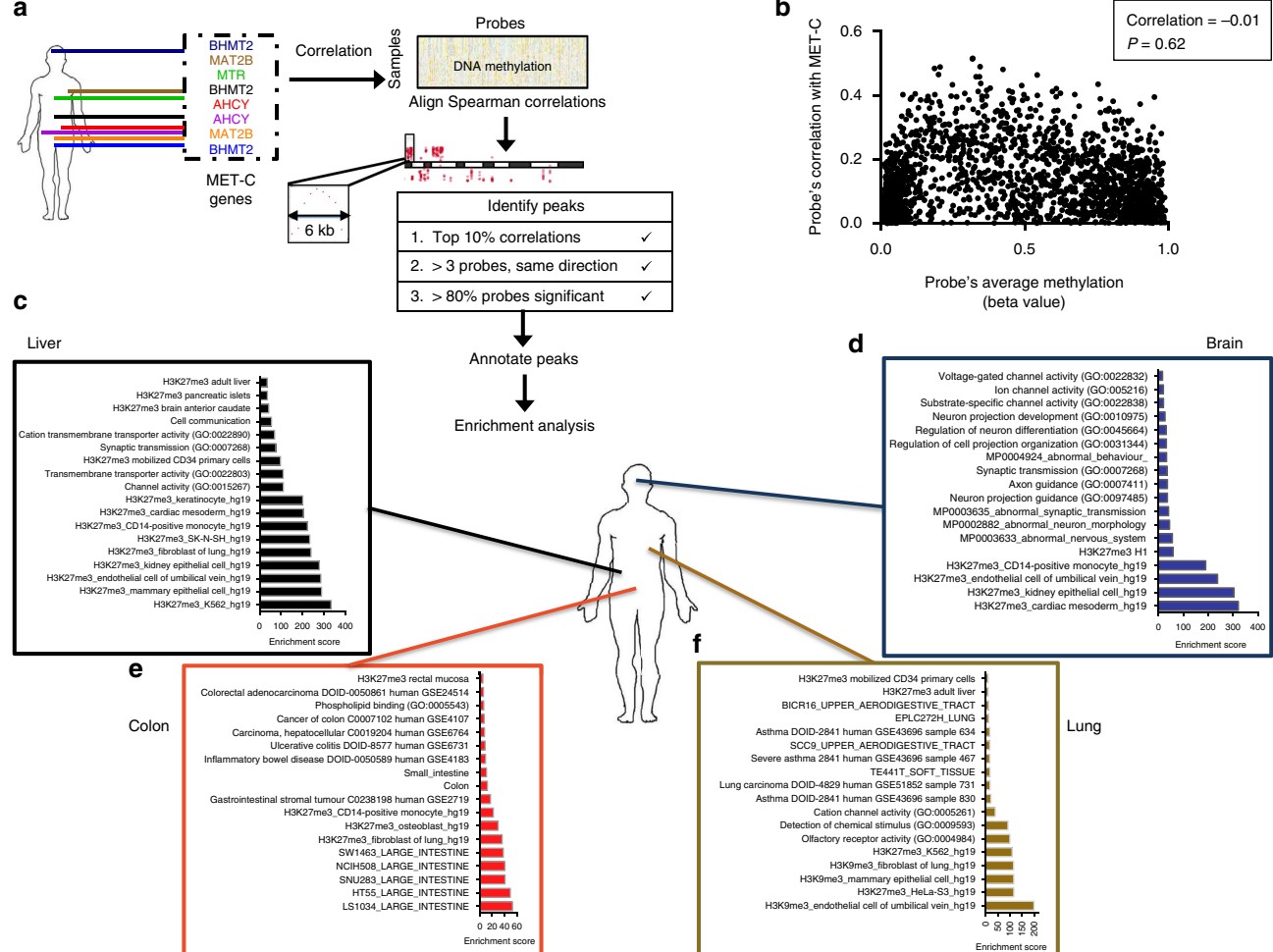

**Figure 3 | Genome-wide screening for metabolically regulated regions.** (**a**) Schematic describing the algorithm used for finding genomic regions where DNA methylation might be regulated by the met cycle (see Methods). (**b**) Assessment of the relationship between met cycle correlation and absolute methylation. The $y$ axis shows the Spearman rho for correlation of 2,000 randomly selected probes with the expression of AHCY in colon cancer. The $x$ axis shows the average methylation level of the same probes across the colon cancer samples in the study (see Methods). (**c**–**f**) Pathway enrichment analyses of genes overlapping peaks. Results are depicted by functional annotation of genes located within peaks of correlation between met cycle and DNA methylation (see Methods for description of gene sets and enrichment scores; see Supplementary Fig. 7 for additional cancer types).

regions consistently across all cancers. Strikingly, many of our peaks overlapped with peaks of histone-3 lysine-27 tri-methylation (H3K27me3) (Fig. 3c–f and Supplementary Fig. 7a–d) as reported by both the encyclopedia of DNA elements (ENCODE) human project[28] and the RoadMap epigenomics project[29]. In cancers of the lung and bladder, histone-3 lysine-9 tri-methylation (H3K9me3) peaks were also significantly enriched (Fig. 3f and Supplementary Fig. 7c). H3K27me3 and H3K9me3 are both associated with repression of gene expression[30]. Our findings therefore suggest that variation in the met cycle's activity may contribute to aberrant expression from normally silenced loci and heterochromatin instability in cancer.

In addition to histone marks, tissue-specific and cell identity gene sets were also enriched in relevant cancer types, including 'breast and ovarian cancer genes' in breast cancer (Supplementary Fig. 7a); 'abnormal nervous system' and 'abnormal neuron morphology' in brain cancer (Fig. 3d); 'asthma' and 'lung carcinoma' gene sets in lung cancer (Fig. 3f); 'kidney-specific' gene set in kidney cancer (Supplementary Fig. 7b); and 'large intestinal genes', 'inflammatory bowel disease' and 'colorectal carcinoma' gene sets in colon cancer (Fig. 3e). Finally, a number of developmental and signalling pathways were among the enriched pathways, including 'transforming growth factor-beta signalling' in kidney (Supplementary Fig. 7b), 'cell communication' pathway in liver (Fig. 3c) and 'G-protein coupled signalling' in bladder cancer (Supplementary Fig. 7c). Organ and embryonic morphogenesis pathways were enriched in breast (Supplementary Fig. 7a), bladder (Supplementary Fig. 7c) and prostate (Supplementary Fig. 7d), all of which are hormonally driven cancers. Interestingly, a previous study in breast cancer showed that embryonic developmental genes are enriched in regions of DNA hypomethylation compared with normal breast[31]. Together, these results illustrate the functional importance of the relationship between met cycle and DNA methylation across cancers.

**Contribution of metabolism to DNA methylation at cancer genes.** So far, we have shown that there is a surprising enrichment of peak regions of metabolically regulated DNA methylation at loci that link to essential aspects of cell identity and chromatin structure. We next questioned whether cancer-specific loci may also exhibit this interaction. We chose 19 well-characterized cancer-related genes such as tumour protein p53 (*TP53*), phosphatase and tensin homologue (*PTEN*) and oestrogen receptor 1 (*ESR1*), as well as 4 genes frequently differentially methylated in cancer, such as APC-WNT signaling pathway regulator adenomatous polyposis coli (*APC*), RAS association domain family member 1 (*RASSF1*), glutathione S-transferase pi 1 (*GSTP1*) and O-6-methylguanine-DNA methyltransferase (*MGMT*)(see Methods). A recent study showed that DNA methylation for any given gene has two major principal components: one representing the promoter region and the other representing the coding sequence[32]. Furthermore, CpG methylation at promoter regions of genes is typically associated with repression, while gene body methylation is thought to increase expression[33]. We therefore applied our integrative modelling to DNA methylation at promoter and gene body regions of each cancer gene separately. In addition to the integrative approach, we also generated models using only the met cycle genes as prediction variables to quantify the predictive ability of met cycle in the absence of other factors. Thus, each cancer gene locus was analysed once using the integrative approach and once using met cycle alone and threefold cross-validation was performed in each case as previously described (Methods). Model performance was evaluated by calculating the error of prediction of test set methylation, as shown for two

examples in Fig. 4: oestrogen receptor 1 (*ESR1*) promoter in breast cancer (MSE = 0.004; Fig. 4a); and androgen receptor (*AR*) promoter in prostate cancer (MSE = 0.001; Fig. 4b). *ESR1* promoter methylation in breast cancer and *AR* promoter methylation in prostate cancer are two examples of events that are known to contribute to the pathogenesis and prognosis of the corresponding tumour types[34–36]. We further assessed the integrative models of promoter methylation at these two loci, and found many SGOC (including met cycle) variables among the top predictive variables of promoter methylation according to the variable importance measures (Fig. 4c,d; Methods).

Notably, the models across all cancers in the study were able to predict cancer gene methylation with high accuracy even using the met cycle variables in the absence of all other variables (85% of the predictions were made with MSE < 0.01; Supplementary Fig. 8a,b). As in the case of local methylation, cancer gene methylation was also more strongly explained by the expression of MAT2B compared with other met cycle variables on average (selected by 24% of all integrative models; Supplementary Fig. 8c), consistent with the function of this enzyme that directly affects SAM levels. Relative variable class comparisons confirmed considerable contribution from the 'methionine cycle enzymes' and 'other SGOC enzymes' among other classes of variables (highest after 'transcription factors' and 'mutations'; Supplementary Fig. 8d,e).

We independently evaluated these findings by applying the same models to both permuted cancer gene methylation values and also randomly generated methylation values (Supplementary Fig. 9a). In all tests, met cycle contribution was significantly (P value < 10e−16) higher when applied to cancer gene methylation versus permutations or random numbers (Methods; Supplementary Fig. 9b), confirming the specificity of signals contained in the true DNA methylation values at cancer loci. Furthermore, we tested the performance of the machine-learning algorithm using randomly generated variables for prediction of cancer gene methylation (Methods) and found in each of the cases tested that the predictions made with the original variables are uniformly more accurate than what is made using simulated random variables (original model MSE smaller by 1.4- to 2-fold than random model MSE on average; Supplementary Fig. 10a–d). We also simulated a data set where prediction variables and the response are related via linear relationships and compared the accuracy of predictions in this simulated linear data set with our original dataset (Methods). We saw in all cases that the improvement in MSE from our data set (MSE 1.4- to 2-fold smaller than random MSE) is even more than what we observed with data of the same dimension that have a linear relationship (MSE 1.3-fold smaller than random MSE; Supplementary Fig. 10e,f). These independent tests confirm that machine learning using the Random Forest algorithm is able to identify non-random signals in the data, and also that it can detect nonlinear relationships between prediction variables and the response.

Next, we ranked all of the variables based on their overall usage according to the integrative models of cancer gene promoter and body methylations (Fig. 4e). Notably, many SGOC (including met cycle) enzymes were among the most frequently selected variables in all cancers (Fig. 4f,g and Supplementary Fig. 11a–f). Importantly, our models highly ranked many clinical and molecular factors previously shown to be associated with DNA methylation in the existing literature (green arrows in Fig. 4f,g and Supplementary Fig. 11a–f). Examples of such positive controls include DNA methyltransferase (DNMT3A or DNMT3B) enzymes[37] that were consistently among the top variables in all cancers (Fig. 4f,g and Supplementary Fig. 11a–f), and patient's age (or age at diagnosis)[12,38] that was highly ranked in prostate, colon, breast, kidney and brain (Fig. 4f,g and Supplementary Fig. 11a–f).

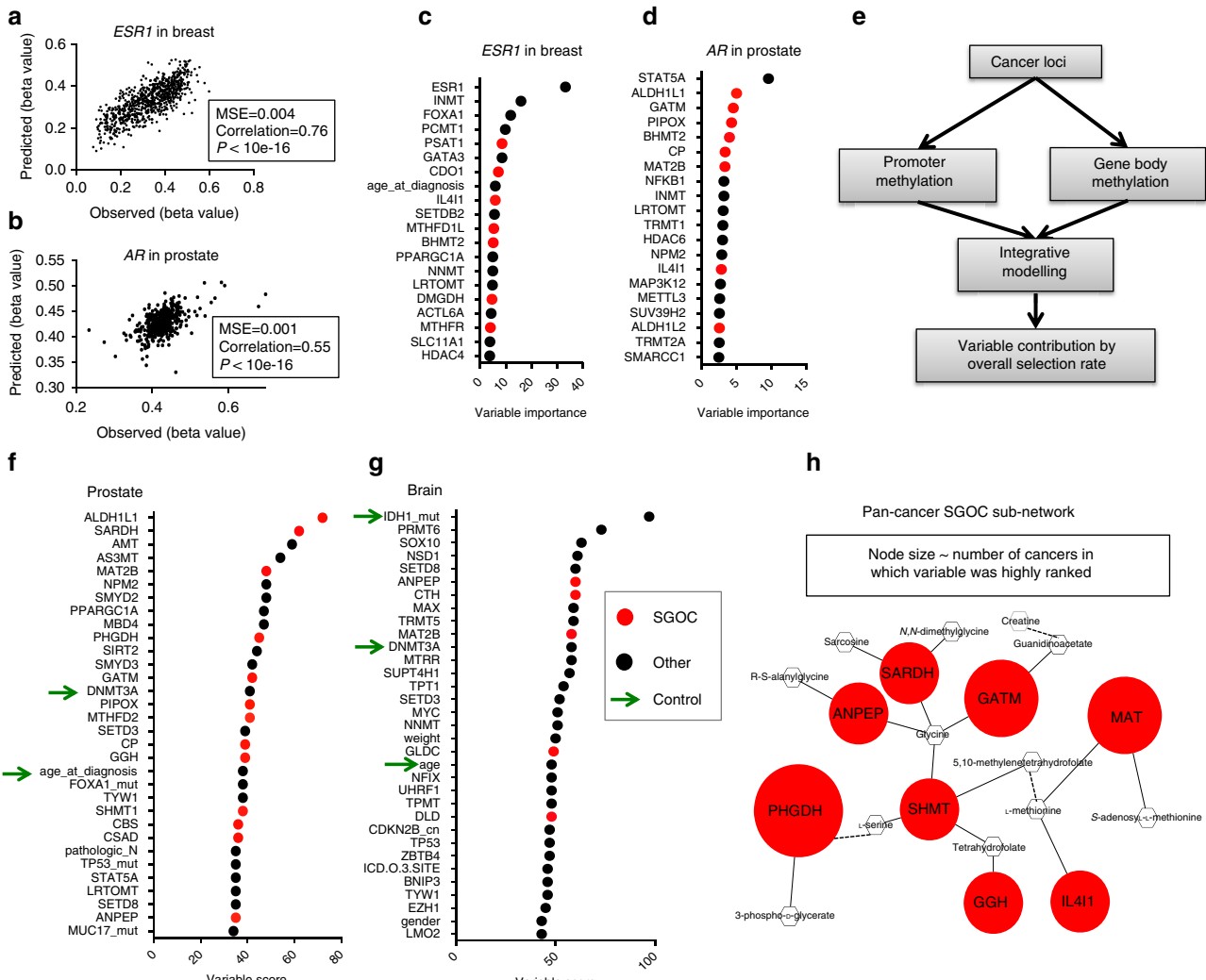

**Figure 4 | Contribution of metabolism to DNA methylation at cancer loci. (a)** Prediction of ESR1 promoter methylation in test samples of breast cancer. The x axis shows the methylation value at ESR1 promoter, while the y axis shows the corresponding predicted values by Elastic Net. **(b)** Prediction of AR promoter methylation in test samples in prostate cancer. The axes are similar to **a**. **(c,d)** Top 20 variables as ranked based on the variable importance score from Random Forest model of ESR1 promoter methylation in breast cancer **(c)** and AR promoter methylation in prostate cancer **(d)**. Variables in the SGOC network (including the met cycle enzymes and other SGOC enzymes) are shown in red and all other variables are shown in black. **(e)** Schematic depicting the ranking of all variables based on combined results of promoter and gene body methylation at cancer loci. **(f,g)** Variables that were most predictive of cancer gene methylation on average (top 15%) are ranked in order of increasing contribution (variable score = per cent usage by Elastic Net). Green arrows point to previously published factors associated with variations in DNA methylation (positive controls). (Variable names: official gene symbols are used to show gene expression variables ('methionine cycle enzymes', 'other SGOC enzymes', 'transcription factors', 'chromatin remodelling factors' and ' SAM metabolizing enzymes'), while '_mut' and '_cn' suffixes following gene symbols denote 'mutations' and 'copy number variations', respectively. For 'clinical factors', variable names match the descriptors used in the TCGA data files.) See Supplementary Fig. 11 for additional cancer types. **(h)** Sub-network of SGOC genes contributing to DNA methylation in multiple cancer types (at least four and three cancers based on Elastic Net and Random Forests models, respectively). Red and white nodes represent genes and metabolite, respectively. Solid edges denote direct biochemical links and dashed edges denote indirect biochemical links through enzymatic reactions not shown. Node sizes for the gene nodes correspond to the number of cancer types wherein each enzyme contributed significantly to cancer gene methylation. (Phosphoglycerate dehydrogenase (PHGDH) = 6, MAT (MAT2B and MAT2A) = 5, glycine amidinotransferase (GATM) = 5, serine hydroxymethyltransferase 1 and 2 (SHMT1 and SHMT2) = 4, sarcosine dehydrogenase (SARDH) = 4, alanyl aminopeptidase (ANPEP) = 4, L-amino acid oxidase (IL4I1) = 4 and gamma-glutamyl hydrolase (GGH) = 4.)

We also observed oestrogen receptor (ER) status to be one of the most important contributors to DNA methylation variation in breast cancer consistent with previous publications[39] (Supplementary Fig. 11b). Furthermore, we found the mutational status of the histone methyltransferase SET-domain containing-2 (*SETD-2*) as a significant contributor in kidney (Supplementary Fig. 11c), smoking in bladder and lung (Supplementary Fig. 11d,f), and isocitrate dehydrogenase 1 (*IDH1*) mutational status in brain cancers (Fig. 4g). Each of these findings are in agreement with the

current knowledge about determinants of DNA methylation[12,40–42]. These results further validate our models and also emphasize the importance of the contribution observed for the SGOC variables (including the met cycle).

Previous work has shown that expression of enzymes across different regions of the SGOC network is predictive of metabolic flux through the network[21]. Notably, we observed that several SGOC genes are consistently among the highly ranked variables by both Random Forest and Elastic Net models in multiple cancer

types. Therefore, to understand which features of SGOC metabolism contribute to the interaction with methylation, we defined a sub-network that was commonly highly ranked by the models in multiple cancer types (Fig. 4h; Methods). This SGOC sub-network comprises the MAT enzymes in the met cycle (MAT2B and MAT2A), as well as enzymes within serine–glycine metabolism such as phosphoglycerate dehydrogenase (PHGDH) and glycine amidinotransferase (GATM) (Fig. 4h). We generally observed negative associations between DNA methylation and expression of PHGDH and GATM, but positive associations with expression of MAT enzymes. A cautionary note however is that in many disease states, levels of particular metabolites in the methionine cycle substantially deviate from physiological ranges, thus activating compensatory mechanism and leading to correlation with DNA methylation in directions opposite of what would be expected from the biochemistry of the reactions[43]. Therefore, when interpreting the direction of correlations between metabolic enzyme levels and DNA methylation, it is important to note that they not only depend on the stoichiometry of the corresponding enzymatic reactions but also on endogenous abundance of the related metabolites. Together, our results suggest that a particular flux configuration through the SGOC metabolic network—which previous studies have shown to be predictable from gene expression patterns[21]—may be important for regulation of DNA methylation.

**Cancer pathogenesis of metabolically regulated DNA methylation.** Involvement of the met cycle in promoter and gene body methylation at cancer genes suggests a potential implication for this metabolic pathway in explaining part of the variability in cancer pathogenesis and patient outcome. To further assess this relationship, we divided patients in each cancer type into two groups based on overall predictability of their cancer loci methylation by the met cycle (see Methods). We then compared survival rates between the two groups ('predictable' by met cycle versus 'not predictable' by met cycle) in each cancer type using the Kaplan–Meier estimator[44] (Fig. 5a–h). An improved overall survival for the 'predictable' group was observed, although the magnitude of this trend varied depending on cancer type with brain, kidney, liver and colon cancers showing statistically significant differences (log-rank test $P$ values: brain $= 3.92e-05$; liver $= 0.0048$; kidney $= 0.0085$; and colon $= 0.04$; Fig. 5a–d). The difference in survival between the predictable and non-predictable groups was not significant in the rest of the cancers studied here (Fig. 5e–h), possibly explained by limited power due to data censoring at later time points. The overall patterns however suggest that the regulation of DNA methylation by the met cycle may be important in maintaining a normal epigenome, and disruption of this relationship in specific subtypes of tumours can lead to high epigenetic stochasticity in those tumours that correspond to poor clinical outcomes. This is consistent with a previous study that showed DNA methylation stochasticity increased across samples with increasing malignancy (from normal to adenoma to carcinoma)[5].

To validate the results of our survival analyses, we applied multivariate Cox regression models to account for covariates such as mutations and clinical factors that are known to be associated with survival rates (Methods). We performed this test in the cases of brain, liver and kidney cancers, where the univariate analyses found highly significant differences between the predictable and non-predictable groups (Fig. 5a–c). The models including covariates still showed a significant difference ($P < 0.05$) between the predictable and non-predictable groups of patients even after taking mutational and clinical factors into account (see Methods for the list of covariates considered in each cancer), suggesting

that a unique part of variation in survival may be explained by epigenetic regulation. We next tried to further validate our results through comparison with independent analyses of the TCGA data by the cBioPortal for Cancer Genomics (cBioPortal)[45] and Prediction of Clinical Outcome from Genomic profiles (PRECOG)[46]. These analyses found lower survival in prostate cancer patients harbouring tumours with deep deletions in the met cycle genes (Supplementary Fig. 12a), and higher survival in kidney cancer patients where the met cycle enzymes are overexpressed (Supplementary Fig. 12b). These results confirm a relationship between met cycle and survival in the same direction as predicted by our hypothesis.

**Discussion**
In this study, we conducted a pan-cancer TCGA analysis of the molecular and clinical contributions to within-cancer (inter-individual) variation in DNA methylation. Through several lines of integrated analysis, we found the overall expression of both the methionine cycle and SGOC network to be strong predictors of multiple aspects of DNA methylation and consistently ranked as one of the highest contributing factors to cancer-associated DNA methylation such as methylation of numerous cancer genes. Within the methionine cycle, we consistently observed a more significant contribution from MAT2B and BHMT2, suggesting that the regulation may be occurring at these enzymatic steps. MAT2B is the enzyme that converts methionine to SAM, therefore it is expected that this enzyme affects SAM levels more directly than other metabolic enzymes. The significance of BHMT2 but not MTR suggests that metabolism of choline and betaine may be more prevalent than folates in cases where one-carbon metabolism fuels DNA methylation. It is important to note that given the nature of our calculations, the results do not prove causal relationships. As such, they should not be interpreted as direct evidence for regulation of DNA methylation by the model variables, but rather as predictive associations.

We introduced a novel approach to identify chromatin regions with strong correlations between DNA methylation and metabolic enzyme levels. The identified regions for the met cycle enzymes significantly overlapped with histone modifications, consistent with enzymatic cross-talk between the two epigenetic processes[30]. The enrichment of gene signatures of repressing histone marks such as H3K27me3 in all cancers points to a possible role for the met cycle in maintenance of DNA methylation at silenced loci. Previous studies have reported aberrant methylation of transcriptionally repressed genes in cancer[47]. In fact, heterochromatin instability arising from increased variability in DNA methylation is a phenomenon observed in many cancers and is thought to contribute to epigenetic plasticity and tumour progression[3,9,48]. Our results provide evidence for this model of dysregulated cancer epigenome and further suggest that disruption of the regulation of DNA methylation by the met cycle—which can be a cause or consequence of tumorigenesis—may be one of the sources of methylation stochasticity leading to higher malignancy. Survival analyses confirm that tumours with a weaker association between their cancer gene methylation and the met cycle expression are more malignant in comparison with tumours wherein this relationship is closer to normal. In addition to epigenetic overlaps, genes with important tissue-specific functions and disease states were also found to fall under the metabolism–DNA methylation peaks. DNA methylation at cell-type-related disease and lineage-specific genes has previously been shown to be dynamic and functionally important[11]. Our results further strengthen the idea that met cycle regulation of methylation is strongly associated with normal tissue function.

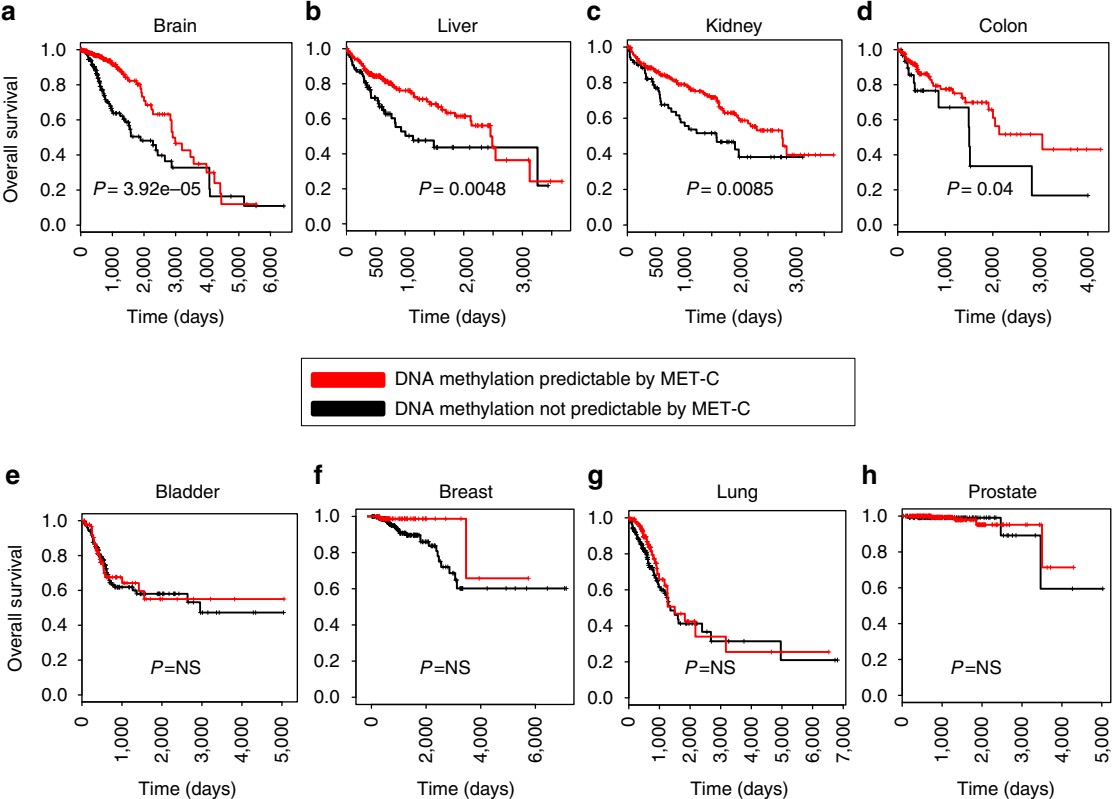

**Figure 5 | Implication of metabolic regulation of methylation in patient survival.** (**a–d**) Kaplan–Meier curves are depicted comparing groups of patients wherein cancer gene methylation was predictable (red) or not predictable (black) by the met cycle variables (see Methods). Overall survival in days is plotted in each case and censored subjects are shown by vertical tick marks (Methods). Log-rank test $P$ value between the two groups is reported. Survival analysis results and log-rank test $P$ values are shown for brain, liver, kidney and colon cancers, respectively. (**e–h**) Survival analysis results as described above are reported for bladder, breast, lung and prostate cancers, respectively. Log-rank test $P$ values showed no significant difference between the 'predictable' and 'not predictable' groups. NS, not significant. Sample sizes: breast = 770; lung = 450; liver = 374; brain = 534; bladder = 408; kidney = 316; prostate = 424; and colon = 198.

Application of the integrative modelling to cancer genes revealed a major role for MAT enzymes (MAT2B and MAT2A), as well as PHGDH and GATM—enzymes involved in serine and glycine metabolism, respectively. Importantly, MAT2B and MAT2A have been shown to co-localize in nuclei and bind DNA through complex formation with chromatin binding proteins providing direct evidence for the role of these enzymes in regulation of transcription via methylation[49]. Our results illustrate that higher levels of MAT2B are associated with more 'regulated' methylation and higher survival, suggesting potentials for genetic or dietary interventions with methionine cycle intermediates in cancer patients. PHGDH diverts the glycolytic flux into the *de novo* serine synthesis pathway that allows glycolysis to provide methyl units. GATM diverts glycine into the creatine synthesis pathway in which SAM is consumed to produce creatine[50]. Creatine synthesis is therefore in competition with the methionine cycle over cellular pools of SAM, explaining why enzymes within the serine–glycine metabolism generally tend to be negatively correlated with the met cycle and DNA methylation.

Overall, this study provides the first comprehensive quantification of the determinants of inter-individual DNA methylation variation in human cancers. The activity of the methionine cycle that emerges in these findings could be either sensed directly by the DNA, or indirectly through interplay with dynamic histone methylation, which itself is tightly regulated by the status of methionine metabolism[18]. Owing to limitation in the coverage of the DNA methylation arrays, it remains to be determined if our

findings are generalizable to methylation across the entire genome, including all non-CpG methylation sites as well as hydroxy-methylation sites. Nevertheless these findings altogether identify metabolism as a major determinant of DNA methylation status in human cancer. It is important to note that the current TCGA data set contains one sample per individual tumour and therefore our conclusions do not necessarily explain the variation in clonal populations within a given tumour. Future studies using multiple samples per tumour or single-cell epigenomics are therefore required to characterize the determinants of intratumour epigenetic heterogeneity. Finally, our study identifies an association between altered tumour metabolism and DNA methylation, while the sources of alterations in metabolism itself remain to be elucidated but can be addressed using similar approaches.

## Methods

**Data curation.** Publically available genome-wide mRNA expression and DNA methylation data were downloaded from TCGA portal (https://tcga-data.nci. nih.gov/tcga/). To increase consistency and minimize unwanted variations, only samples processed using RNASEQ-V2 with level 3 gene-normalized RNA-seq by expectation maximization values for gene expression, and level 3 beta values from Illumina Infinium HumanMethylation450K BeadChip data for DNA methylation were included in the study. We selected the following eight cancer types wherein the number of available samples analysed on both platforms was sufficiently large for machine-learning calculations: 770 samples of breast invasive carcinoma (BRCA); 450 samples of lung adenocarcinoma; 374 samples of liver hepatocellular carcinoma; 534 samples of brain lower-grade glioma; 408 samples of bladder urothelial carcinoma; 316 samples of kidney renal clear cell carcinoma; 424 samples of prostate adenocarcinoma; and 198 samples of colon adenocarcinoma. Somatic

mutations with a frequency of 5% or higher, and Genomic Identification of Significant Targets in Cancer (GISTIC) values for copy number alterations with a frequency of 15% or higher according to the cBioPortal[45] were obtained and included in the models. Clinical and follow-up data were downloaded via the TCGA-Assembler[51].

**Assessment of batch effects.** We used the TCGA Batch Effects online tool (http://bioinformatics.mdanderson.org/tcgabatcheffects) to check for the existence of batch effects in the data used in our study. For each cancer types in our study, both the DNA methylation and the RNA-seq batch effects were negligible (dispersion separability criterion score <0.5 for all sample batches included in the study).

**DNA methylation.** The Illumina Infinium Human Methylation450K BeadChip consists of more than 450,000 probes across the genome covering CpG sites within and outside of CpG islands as well as non-CpG methylation sites identified in embryonic stem cells (http://www.illumina.com/products/methylation_450_beadchip_kits.html). We first filtered all probes with more than 80% missing values across each cancer type. Global DNA methylation was then defined as the average beta value across all remaining probes for each sample (Supplementary Fig. 1a). Sex chromosomes were also excluded from all subsequent analyses of DNA methylation. To assess local DNA methylation, we divided the genome into 10 kb intervals and calculated the average beta value across all probes within each bin. We then filtered regions where variation in methylation was modest (s.d. <0.2 across each data set). The average beta value across all remaining 10 kb regions was then calculated for each sample individually and plotted in Supplementary Fig. 1c. To study DNA methylation at cancer loci, probes that mapped to each gene according to Illumina annotations were identified. Promoter DNA methylation was then defined as the average beta value across all probes mapping to a given gene and falling within one of the following positional categories based on Illumina chip annotation information: 'TSS1500'; 'TSS200'; or '5′UTR'. Gene body methylation for each gene was defined as the average beta value across all probes mapping to a given gene and falling in '1st exon', 'Body' or '3′UTR' based on the annotation. Promoter and gene body methylation were separately modelled for each of the cancer genes in the study (Fig. 4).

**Gene expression.** Log-transformed gene-normalized RNA-seq by expectation maximization values were used as expression levels, and low-expression genes in each data set were defined as having <70% of the samples with a count value larger than 3. Such genes were removed from further analysis.

**Gene expression variables included in the integrative models.** In addition to the major enzymes in the met cycle (MAT2B, MTR, BHMT2 and AHCY), four classes of expression variables with potential links to DNA methylation were also included in the integrative models (see Supplementary Data 1 for the complete list of variables). The four classes are described in the following sections.

*Other SGOC enzymes.* SGOC metabolic genes from our previous network reconstruction were included[21]. To separately assess the effect of the met cycle from the rest of the network, we excluded the met cycle enzymes from this class and treated them as a separate class (methionine cycle enzymes).

*Chromatin remodellers.* A list of human chromatin remodellers and DNA methylation machinery was constructed by combining the Gene Ontology (GO) chromatin modifiers list, GO chromatin remodellers list[52], and methylated DNA-binding proteins and de-methylases[53].

*Transcription factors.* For each cancer type, transcription factors important in the pathogenesis or subtype specification based on previous literature were included[54].

*SAM metabolizing enzymes.* DNA methyltransferases and other SAM-consuming enzymes (except for MAT enzymes already included in the class 'methionine cycle enzymes') according to Human Cyc[55] were included in this class.

**Mutations included in the integrative models.** For each cancer type, genes with frequent somatic mutations (minimum frequency of 5%) among the TCGA cohort according to the cBioPortal[45] summary table (TCGA, Provisional) were obtained. The transposed matrix of individual barcodes and mutations in the selected genes was downloaded from the cBioPortal for each of the eight cancers in this study. See Supplementary Data 1 for a complete list of somatic mutations considered in each cancer type.

**Copy number alterations included in the integrative models.** For each cancer type, genes with frequent copy number alterations (minimum frequency of 15%) among the TCGA cohort according to the cBioPortal[45] summary table (TCGA, Provisional) were obtained. The transposed matrix of individual barcodes and putative copy number alteration calls by GISTIC[56] for the selected genes was downloaded from the cBioPortal for each of the eight cancers in this study (values of putative copy number calls determined using GISTIC 2.0: −2 = homozygous deletion; −1 = hemizygous deletion; 0 = neutral/no change; 1 = gain;

2 = high-level amplification). See Supplementary Data 1 for a complete list of copy number alterations considered in each cancer type.

**Clinical factors included in the integrative models.** For each cancer type, clinical information was downloaded through the TCGA-Assembler[51]. All clinical attributes were included for each cancer type with the exception of the ones filtered out due to missing data for all samples or factors with the same level across all samples. See Supplementary Data 1 for a complete list of clinical attributes considered in each cancer type.

**Variable ranking using the Random Forest algorithm.** The Random Forest is a machine-learning algorithm that generates predictions by averaging over a collection of randomized decision trees. Since successive trees are built with bootstrap samples, the algorithm is robust to over-fitting, and also those samples that are left out (the out-of-bag samples) can be used to quantify the contribution that prediction variables make to the overall response. The Random Forest method is designed to accommodate nonlinearities between the response and prediction variables, as well as unknown interactions among the variables[57,58]. We used the R package 'randomForest'[59] and performed threefold cross-validation by manually dividing the samples in each cancer type into three training and test subsets. To build each forest, tree size was set to 500 and the 'importance' parameter was set to 'TRUE' in the R function 'randomForest' so as to provide estimates for the importance of prediction variables. Missing data were imputed using the 'na.roughfix' function in the 'randomForest' package. We obtained separate measures of importance for each variable from each Random Forest run. These importance scores are calculated as the per cent increase in the mean squared prediction error on the out-of-bag samples when a given variable is permuted. Variables were ranked based on average importance scores across all cross-validation folds. Prediction errors were calculated as the mean squared difference between the predicted versus the observed methylation values for the test set samples. The square root of the MSE has the same scale as the response (DNA methylation beta values in this case), and is therefore a direct measure of the accuracy with which predictions were made. (Fig. 1c and Supplementary Fig. 8a,b).

**Variable selection using the Elastic Net algorithm.** Elastic Net is a penalized regression approach for variable selection and quantitative inference that identifies linear combinations of unique variables that contribute to a response variable such as the amount of DNA methylation. The algorithm was developed and benchmarked to avoid over-fitting in statistical modelling of high-dimensional data containing collinearity[60]. We applied the Elastic Net algorithm using the R package 'glmnet'[61]. Elastic Net performs variable selection by minimizing a regularized cost function using the following equation

$$\min_{\beta 0, \beta} \frac{1}{N} \sum_{i=1}^{N} \mathbf{w}_i l \left( \mathbf{y}_i \beta 0 + \beta^T \mathbf{x}_i \right) + \lambda \left[ (1 - \alpha) \parallel \beta \parallel_2^2 / 2 + \alpha \parallel \beta \parallel_1 \right]$$

where $\lambda$ is the tuning parameter and $\alpha$ is the Elastic Net penalty term. For each cancer type, the samples were divided into three independent test subsets (threefold cross-validation), and separate models were generated using each training subset. Using a grid of different tuning parameter values, we found the $\lambda$ that minimized the MSE using fivefold cross-validation within each training set for each model separately. The value of $\alpha$ was set to 0.5 to handle potential correlated variables. Finally, for each variable, average coefficient across the three independent models was calculated for each region and each cancer type. Owing to the existence of categorical factors among our variables (for which scaling is not appropriate), we also calculated the selection rate as an alternative measure of variable importance referred to as 'variable usage' in the manuscript. Variable usage was measured as the fraction of times across all cross-validation folds that a variable was selected by the Elastic Net to be included in the final model (Supplementary Figs 8c and 11a–f and Fig. 4f,g). Finally, prediction errors were calculated as the squared difference (MSE) between the predicted and measured DNA methylation values for the test sets (Fig. 1c and Supplementary Fig. 8a,b).

**Variable class contributions to DNA methylation.** Variables were functionally categorized into the following eight classes: 'methionine cycle enzymes'; 'other SGOC enzymes'; 'chromatin remodelling factors'; 'transcription factors'; 'SAM metabolizing enzymes'; 'clinical factors', 'copy number variations'; and 'mutations'. Results of the integrative modelling were summarized and reported in terms of the average contribution from each of the above functional classes in explaining DNA methylation variation. Variable importance scores from Random Forest models were averaged across all variables within a given class, and an overall class importance score was calculated. In the case of Elastic Net models, variable usage as described in the previous section, was averaged across variables in each class and an average percentage showing selection rate was calculated. Finally, classes were ranked in each cancer type according to their average contribution and the overall class ranks were plotted in Fig. 2a,b and Supplementary Figs 4a,b and 8d,e.

**Comparison with gene expression controls.** A set of 100 randomly selected genes from the genome with similar cross-sample variation in expression as our

original gene expression variables (transcription factors (TFs), SGOC, MET-C, SAM and chromatin remodelling factors (RMs)) were considered. We performed this test on local DNA methylations (all variable 10 kb regions) in brain cancer (lower-grade glioma) as an example and repeated the integrative modelling using this set of randomly selected genes in addition to all other variables present in the original models. All gene expression variables were then ranked using a similar approach as described above. To compare our original gene expression variables with the variance-matched random genes, the ranks across all models were averaged (Fig. 1d), and P values were obtained from one-tailed Mann–Whitney non-parametric test between the two groups from Elastic Net and Random Forest. To further test our gene expression variables against other gene families, five popular gene sets were considered: RTK, RSK, TLR, MAPK signalling and WNT signalling families. The list of genes in these families were obtained from the HUGO Gene Nomenclature Committee (http://www.genenames.org/cgi-bin/genefamilies/). The same approach as described above for randomly selected genes was used to compare these gene sets with our original gene expression variables (Supplementary Fig. 3).

**Distance to nearest gene TSS.** Selected 10 kb regions were converted to genomic range objects using the R package 'GenomicRanges'[62]. The distance to single nearest gene's TSS was found using Genomic Regions Enrichment of Annotations Tools (GREAT)[63]. Genomic regions are associated with nearby genes by first assigning a regulatory domain to every gene in the genome, and then finding genes whose regulatory domains overlap with a given genomic region. We set the association rule parameter in GREAT to 'single nearest gene' with a maximum extension of 1,000 kb for definition of regulatory domains. Density plots of distance to TSS are depicted in Supplementary Fig. 2b. The same approach was used for annotating peaks obtained from Fig. 3 (density plots shown in Supplementary Fig. 5a,c). To obtain the distribution of Illumina probe densities around the TSS, we randomly selected 10,000 probes across the arrays and applied the above-described approach to measure the distance to nearest gene's TSS for each probe. Density plots were obtained for the purpose of comparison with the distribution of metabolically regulated peaks (Supplementary Fig. 5b,d).

**Identification of metabolically regulated genomic regions.** To find peaks of strong association between the met cycle and DNA methylation, we designed a novel scanning method by applying the idea of Manhattan plots from expression quantitative trait loci analyses to DNA methylation data. In each cancer type, we first selected one of the major enzymes in the met cycle with the highest overall Spearman correlation with global and local DNA methylations (BHMT2 in brain, breast, prostate and liver; MAT2B in lung and bladder; and AHCY in colon and kidney cancers), and calculated the Spearman correlation between its expression and the beta value of each individual probe across the genome. We then sorted the probes according to genomic coordinates and aligned the $-\log10$ of the P values obtained from the Spearman correlations along the chromosomes. Next, we applied a sliding window scan for regions of strong association across the genome separately in each cancer type (Fig. 3a). For this, probes with the highest correlations (top 10% across the genome) were located and a 6 kb window ($+3$ and $-3$ kb) flanking the genomic coordinate of the original probe was scanned. A region was reported as a 'peak' if the following criteria were met: (1) region included at least three probes with a correlation in the same direction as the original probe (positive or negative); (2) at least 80% of all probes within the region had a significant ($P < 0.00001$) correlations with met cycle expression. After applying these filters, the selected regions were annotated and genes overlapping with each of the peaks were used for subsequent pathway enrichment analyses. Given the window size and the above criteria, the majority of the identified peaks only overlapped with one unique gene (see Supplementary Data 2 for a complete list of all identified peaks).

To assess potential bias towards highly methylated regions in the identified regions where correlation of methylation with met cycle expression peaks, we tested 2,000 randomly selected probes across the genome. We then evaluated the association between methylation of each probe with the value of its Spearman correlation rho with met cycle expression—we used AHCY in colon cancer as an example in this test (Fig. 3b).

Finally, an additional filter was applied to rank the identified peaks according to peak shape. For this, the aligned correlation coefficients in each region were assessed with respect to whether they formed a peak according to an information theory score calculated by the R function 'turnpoints' (refer to R package 'pastecs'[64]). This function finds all turning points (peaks and pits) in a series of points (in this case, aligned correlation coefficients), and calculates the information quantity of each turning point using Kendall's information theory. Finally, it measures a P value against a random distribution of the turning points in a given series, with smaller P values corresponding to less random shape and a higher probability of a turning point corresponding to a real peak or pit. We selected regions containing turning points with the most significant P values (lowest 20%) in each cancer type and subsequently tested them for specificity for the met cycle as described in the following section.

**Test of specificity of peaks for the met cycle.** Each of the selected peaks was tested for specificity of their correlations with the met cycle expression (versus gene expression in general). For this, 500 genes were randomly selected from the genome in each cancer type, and the Spearman correlation coefficient was measured between their expression and the methylation of every probe within a given peak. The fraction of significant correlations was calculated for all of the 500 genes as well as for the met cycle gene. A randomization q-value was calculated for the met cycle gene by comparing it with the distribution of the correlations calculated for the 500 random genes. This procedure was repeated separately for each peak in each cancer type and the results are summarized in Supplementary Fig. 8a,b.

**Pathway enrichment analyses.** Peaks were annotated according to Illumina information and UCSC Ref gene names for genes overlapping with the identified peaks were extracted. Pathway enrichment analysis was performed on the resulting gene list for each cancer type using Enrichr[27]. Combined scores from Enrichr were used to rank pathways. The combined score 'c' is defined as $c = \log(p) \times z$, where p refers to the P value from the Fisher's exact test and z is the z-score indicating the deviation from the expected rank. Enrichr first calculates Fisher's exact P values for many random gene sets to generate a distribution of expected P values for each pathway in their pathway library. The z-score for deviation from this expected rank is therefore an alternative ranking score and the combined score is considered a corrected form of the enrichment score and P value, which we used to sort pathways in Fig. 3c–f and Supplementary Fig. 7a–d. All gene sets in Fig. 3c–f and Supplementary Fig. 7 had Fisher's exact P values $< 0.05$, and the most highly enriched sets are shown ranked by the combined enrichment scores. Gene set names used in Fig. 3c–f and Supplementary Fig. 7 follow the convention used and described by Enrichr (http://amp.pharm.mssm.edu/Enrichr/#stats). Briefly, GO sets are shown by GO numbers in parenthesis following their name, epigenetic modifications from the ENCODE histone modifications 2015 project are shown by '-hg19' following gene set names to be distinguished from those from the Epigenomics Roadmap project, gene sets from the Cancer Cell line Encyclopedia are shown by cell line names following cancer type in upper case, disease signatures from the gene expression omnibus are shown in upper case followed by GSE accession numbers, KEGG 2015 and the Human Gene Atlas gene sets are shown in lower case. Refer to Enrichr for a complete list of all gene sets included in more than 70 libraries.

**Cancer genes.** A list of 12 cancer drivers common in multiple human cancers was considered[65] (TP53, PTEN, neuroblastoma RAS viral oncogene homologue (NRAS), EGFR, IDH1, IDH2, CCCTC-binding factor (CTCF), von Hippel–Lindau tumour suppressor, E3 ubiquitin protein ligase (VHL), catenin beta 1 (CTNNB1), nuclear factor erythroid-2 like 2 (NFE2L2), phosphoinositide-3-kinase, regulatory subunit 1 (PIK3R1) and ms-related tyrosine kinase 3 (FLT3)). These genes were consistently identified as candidate cancer drivers by four independent positive selection detection algorithms in a comprehensive pan-cancer analysis of thousands of TCGA tumours[65]. We added to this list, well-known cancer drivers not included in the above list (Kirsten rat sarcoma viral oncogene homolog (KRAS), B-Raf proto-oncogene, serine/threonine kinase (BRAF), phosphatidylinositol-4,5-bisphosphate 3-kinase, catalytic subunit alpha (PIK3CA), and breast cancer 1, early onset (BRCA1)). In addition to these common cancer drivers, we also considered a number of cancer type-specific genes, including receptors important in specific subtypes of cancers (ESR1, AR and erb-b2 receptor tyrosine kinase 2 (ERBB2)). Finally, cancer genes frequently aberrantly methylated in human cancers were also considered[34] (RASSF1, GSTP1, APC and MGMT), together constructing a list of 23 cancer genes for detailed analysis of DNA methylation shown in Fig. 4.

**Evaluation of model performance using randomized responses.** To test the reliability of the variable contribution results obtained from our gene-specific DNA methylation models, we built two different randomized data sets as control responses, each with the same dimensions as the original response data set (that is, the cancer gene DNA methylations). In the first case, we permuted the DNA methylation values of each cancer gene, and repeated the modelling using the met cycle variables. In the second case, we generated random beta values (from uniform distribution in the range of 0–1) and used those as the response variables in the calculations. We then compared average met cycle variable importance (Random Forest) and variable usage (Elastic Net) from prediction of true cancer gene methylations versus permuted methylations and randomly generated responses. The Kolmogrov–Smirnov test P values were calculated between the distributions as illustrated in Supplementary Fig. 9b.

**Evaluation of model performance using randomized predictors.** Using prostate cancer as an example, we performed simulation tests to determine whether the Random Forest as a methodology is able to utilize the information in the prediction variables beyond what could be expected if the predictors were only random noise and unrelated to the response. To investigate this, we modelled methylation in the prostate cancer data set at three example cancer loci (GSTP1, RASSF1 and PITX2). These genes were selected based on previous evidence indicating the critical

importance of their aberrant methylation in prostate cancer[34,66]. As controls, we generated three additional data sets. For the first data set, we copied the exact response as the *GSTP1* methylation, but randomly generated a predictor variable set of the same dimensions as the original variable set by sampling from a standard normal distribution. That is, each observation on each variable is a sample from a normal distribution of unit variance and should therefore have no relationship to the response. The other two data sets were generated in the same manner, using *RASSF1* and *PITX2* methylation as responses and randomly generated variable sets as predictors. To assess the performance of the Random Forest computations, we compared the MSE from predictions made using the original data with those made by the data sets consisting of random variables unrelated to the responses. For each of the three responses, we randomly divided the data into training and test sets, generated a total of 100 simulations consisting of 500 decision trees, and compared the resulting MSEs of the predictions made on the test points. Results are summarized in Supplementary Fig. 10a–d.

To quantify the improvement in the Random Forest algorithm by using the original variables over the randomly simulated variables, we defined an improvement metric (MSE-Imp), describing the relative improvement in prediction accuracy:

$$\text{MSE-Imp} := \frac{\text{MSE-rand}}{\text{MSE-orig}}$$

where MSE-rand is the average MSE calculated using the random simulated variables and MSE-orig is the average MSE calculated using the original variables.

In this test, another simulated data set of the same dimensions as the original data set was generated where the variables and response were linearly related via the following equation:

$$Y = \sum_{i=1}^{P} \boldsymbol{\beta}_i \mathbf{X}_i + \varepsilon$$

To generate this linear data set, we sampled the value of each prediction variable $\mathbf{X}_i$ from a standard normal distribution and the noise $\varepsilon$ from a normal distribution with mean 0 and s.d. 0.05. The values of the coefficients $\boldsymbol{\beta}_i$ were selected uniformly at random from the interval [0,1]. We then measured the MSE improvement (MSE-Imp) for the linear data set using the same approach as MSE improvements for the original data sets were calculated (explained in the previous paragraph). This allowed us to compare a linearly simulated data set with our real data set. Results are shown in Supplementary Fig. 10e,f.

**Network construction.** Genes in the SGOC network (including the met cycle genes) that were among the most highly ranked variables (top 15%) in at least four of the cancer data sets according to the Elastic Net models and at least three of the cancer data sets according to the Random Forest models were selected. A metabolic network consisting of these enzymes was then constructed using MetScape[67], where nodes represent genes and metabolites, and edges represent biochemical links. We fixed the node size for metabolites but adjusted node sizes for genes to correspond to the number of cancers in which each variable was highly ranked (among the top 15% of all variables) (Fig. 4g). For nodes not directly connected to the rest of the network, we manually added dashed lines where appropriate.

**Survival analyses.** In each cancer type, the average error of prediction of DNA methylation at cancer loci was measured for each patient across all Elastic Net models using only met cycle variables for prediction. Patients were then divided into two groups based on predictability of their methylation by the met cycle activity ('predictable' = below-median prediction error; 'not predictable' = above-median prediction error). To estimate overall survival time, 'days-to-death' was used with vital status information and last follow-up date used to right-censor subjects (subjects alive at last follow-up were censored from the analysis beyond their last follow-up date). The relationship between survival and predictability was then analysed using the 'survfit' function in the R package 'survival'[68] and visualized by Kaplan–Meier curves. Log-rank test *P* values were calculated by fitting models of overall survival to patients' 'predictability' group assignments using the 'survdiff' function in the survival package for each cancer type separately. Results are depicted in Fig. 5.

**Multivariate Cox regression.** In the three cancer types (brain, liver and kidney), where univariate analysis showed a highly significant difference in survival between the predictable and non-predictable groups as described above, and also the sample size allowed for sufficient power to perform multivariate analysis, we used relevant clinical and mutational factors as covariates and repeated the survival analysis. The following factors were individually tested as covariates in separate models of overall survival along with 'predictability' status as the fixed effect: brian cancer: all frequent somatic mutations (see Supplementary Data 1 for the complete list), histological diagnosis, age, gender and initial weight; liver cancer: all frequent somatic mutations (see Supplementary Data 1 for the complete list), tumour stage, history of other malignancies and residual tumour; kidney cancer: all frequent somatic mutations (see Supplementary Data 1 for the complete list), age and race. In each case, the results of regression using the 'coxph()' function in R provided the

*P* value for the significance of the predictability status when modelling overall survival in the presence of covariates.

**Software.** All computational and statistical analyses were done using R 3.1.2 (ref. 69). Distribution plots, box plots, scatter plots and bar plots were made in GraphPad Prism version 6 (GraphPad Software, San Diego California USA, www.graphpad.com). Circular plots were generated using Circos[70].

**Code availability.** R script is available through the following Github repository (https://github.com/mahyam/DNA-methylation-and-metabolism-R-code).

**Data availability.** All data used in this study were obtained from the TCGA data portal available online at https://gdc.nci.nih.gov/.

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

## Acknowledgements

We thank Sam Mentch for useful comments and help with the graphics. This work was supported by grants R00CA168997 and R01CA193256 to J.W.L.; and T32GM007617 and F99CA212457 to M.M. from the National Institutes of Health. M.M. was also supported by a Graduate Fellowship from the Duke University School of Medicine.

## Author contributions

M.M. and J.W.L. designed the study; M.M. and J.W.L. wrote the manuscript with essential comments from A.G.C.; M.M. performed the analyses; L.K.M. performed the simulation tests for evaluation of the Random Forest models; J.W.L. supervised the project.

## Additional information

**Competing financial interests:** The authors declare no competing financial interests.

