## [Peer Review File · Nature Communications]

Reviewers' comments:

Reviewer #1 (Remarks to the Author):

In this paper, the authors developed integrative models to identify determinants of DNA methylation variation. The authors identified that a set of metabolic genes involved in the methionine cycle and the serine glycine pathways are predictive of several features of DNA methylation status in tumors including the methylation of genes that are known to drive oncogenesis. The authors claims that their study is the first comprehensive analysis of the determinants of methylation and demonstrates the surprisingly large contribution of metabolism in explaining epigenetic variation among tumors. Some of the analyses are results are quite interesting. But I also found some of the results presented here seem selected without much justification. I have the following questions about their analysis method and their conclusion.

Major:

1. I found the first part of the results, "pan-cancer analysis of global and local DNA methylation" has little to do with the main theme of the paper. I found "Notably, as liver is the main metabolic organ, highest level of variability in DNA methylation observed in liver cancer suggests that disrupted metabolic mechanisms may be involved in loss of epigenetic stability." (lines 130-133) a bit far-fetched and speculative.
2. The factors selected to be used in the integrative model seems highly selective. Only the "highly-related" ones are included. I wonder whether the authors can include some control variables, ones that are not known to be associated with methylation change, to serve as controls.
3. Is elastic net applied also when only the Met-C variables is used? Not sure if this is necessary when there are only 4 variables?
4. I am not sure about how good the fit is for the all the models discussed here. Is MSE 0.08 considered a good fit?
5. Lines 190-191. The authors claimed that "MAT2B and BHMT2 exhibit a larger contribution to local DNA methylation among the met cycle variables". However, in two cancer types, Colon cancer and Bladder cancer, the BHMT2 variable was selected 0% as shown in both Figures 2d and 2e.
6. Lines 209-213, "Results from both Random Forest and Elastic Net algorithms confirmed a considerable contribution from the class "Methionine cycle enzymes" relative to other classes of variables (3rd strongest scoring among 8 classes, closely following "Transcription Factors" and "Other SGOC enzymes") according to both Elastic Net and Random Forest models (Figs. 3b,c)." I am not sure where does the rank comes from. But it seems to me in Figure 3C, Met-C is ranked 4th behind TFs, SGOC and MUTs! I am confused.
7. Line 911. "All gene sets in Fig. 4 had Fisher's exact p-values <0.5". Is this a typo? If not, I would expect to see a very long list of pathway names in theses figures.

Minor:

What are the references at the end (lines 1040 and beyond) intended for?

Reviewer #2 (Remarks to the Author):

The idea of this paper is very interesting, and one must give a lot of credit to the authors for attempting to tease out potential mechanistic associations of what would be trans-interactions of genes regulating methylation, to the methylation changes in cancer. The problem(s) is that the paper is extremely confusing, which is partly because of the design, partly the writing, and partly the fact they are overlooking some key principles of cancer genetics as well as relevant literature. Criticisms are not in any particular order here:

1. They would have to be extremely careful about arguing for mechanism. One cannot infer cause and effect from a study such as this. The very fact they are using machine learning means they are not defining a mechanistic link between one event and another, i.e. they don't know the causal flow here.
2. There is only one sample per tumor so they can't address intratumoral heterogeneity. At this point relating variability from person to person is fairly removed from real tumor heterogeneity, since this could be genetically driven.
3. It is not an agnostic study in that they set out with a group of genes they think might be important. They don't compare to a similar "important group", like tyrosine kinases, or signaling receptors, etc.
4. 200 variables seems overspecified. Plus there are 18,000 models! How do you correct for this large number? The data in Figs 3 and 4 look highly overspecified.
5. The methionine cycle itself is highly sensitive to metabolic changes in cancer, which we know are very much disturbed. So how do you know that the methylation changes and the methionine cycle aren't both disturbed by another extrinsic metabolic factor?
6. They do not make any tumor-normal comparisons.
7. They do not address batch effects and RNA quality in the TCGA data. These are potentially huge problems that could underlie their results.
8. They do not address cell type heterogeneity, which is really surprising given the level of statistical expertise on the paper. In particular lymphocyte infiltration would likely cause big changes in methylation metabolism (they sue the Tet system, for example).
9. Could simply be degradation related, which also induces biochemical changes. What about cell type heterogeneity? This could have a major effect on everything they are talking about. Immune cells when activated are highly dynamic metabolically and also are known to use the Tet system.
10. They use bead chip only. That is very confusing since there are scattered CpG sites on these arrays. They also didn't use a block finder.
11. The 10 kb bin they use is highly biased toward what was curated in the first place, i.e. highly functional regions of interest to 22 contributing labs, including DNase sensitivity, regulation etc., and is also enriched in CG dense regions.
12. As far as I can tell there is no replication set, e.g. working out the model in one group of tumors and checking in a different one. They just do out-of-bag calculations to quantify predictions. We know a lot about huge methylation changes in glioma and leukemias due to mutations in specific methylation pathway genes. They don't start with that against known relevant data sets, and test that against papers that already address these issues, then go back and see if this same yardstick makes sense in the other tumor types.
13. They do not seem to be examining mutational status at all! One would then want to do a multivariate analysis against known mutations, and against other survival indicators.

May 23, 2016

Reviewers' comments:

Reviewer #1 (Remarks to the Author):

In this paper, the authors developed integrative models to identify determinants of DNA methylation variation. The authors identified that a set of metabolic genes involved in the methionine cycle and the serine glycine pathways are predictive of several features of DNA methylation status in tumors including the methylation of genes that are known to drive oncogenesis. The authors claims that their study is the first comprehensive analysis of the determinants of methylation and demonstrates the surprisingly large contribution of metabolism in explaining epigenetic variation among tumors. Some of the analyses are results are quite interesting. But I also found some of the results presented here seem selected without much justification. I have the following questions about their analysis method and their conclusion.

We thank the reviewer for their time and positive evaluation of our study. We also appreciate the criticisms the reviewer notes. We have worked to provide further clarification of the justification of our results.

Major:

1. I found the first part of the results, "pan-cancer analysis of global and local DNA methylation" has little to do with the main theme of the paper. I found "Notably, as liver is the main metabolic organ, highest level of variability in DNA methylation observed in liver cancer suggests that disrupted metabolic mechanisms may be involved in loss of epigenetic stability." (lines 130-133) a bit far-fetched and speculative.

We thank the reviewer for mentioning this point and apologize for the confusion this figure brought about. Our original intention was that Figure 1 would provide a description of the overall variation in DNA methylation across the samples used in this study so that the data could be introduced and serve as a starting point for the integrative modeling. Nevertheless in hindsight we suspect this figure may have distracted from the key points we make in the study.

Therefore, we have moved this analysis to a supplementary section of the paper in the revised version. We have also removed the above-mentioned statement about liver methylation to address the reviewer's concern and remove that speculative mention.

2. The factors selected to be used in the integrative model seems highly selective. Only the "highly-related" ones are included. I wonder whether the authors can include some control variables, ones that are not known to be associated with methylation change, to serve as controls.

This, we believe, is a critical point the reviewer raises. We began to address this essential point mainly in Supplementary Figure 7 in which we used randomized variables and showed that the models with selected variables are able to distinguish between noise and the true associations.

We have further addressed this point by including an additional group of randomly selected genes = thought to be unrelated to methylation in our models and repeated the analyses to improve confidence in the performance and results of the models (Revised Figure 1d).

Another point we have better emphasized in the revised version of this manuscript is that the key finding of our study is that the methionine cycle and other one-carbon metabolism related variables perform better at predicting methylation than most known variables that are thought to determine DNA methylation status.

3. Is elastic net applied also when only the Met-C variables is used? Not sure if this is necessary when there are only 4 variables?

Yes, in the interest of technical consistency across all models, we used the same algorithms for variable selection in all cases (Elastic Net and Random Forest), though as the reviewer correctly points out, this was not necessary for models with small number of variables. Nevertheless, the Elastic Net is a generalized linear model and is more suitable than a standard regression model when small numbers of potentially correlated variables are used.

4. I am not sure about how good the fit is for the all the models discussed here. Is MSE 0.08 considered a good fit?

We chose to report MSE as the measure of the goodness of fit since it is the most widely used metric for comparing model performances in both Elastic Net and Random Forest methodologies in both cases when the dependent variable is continuous¹⁻³. The advantage to reporting the MSE is that the square root of this value has the same scale as the dependent variable and provides an absolute

measure of the predictive capacity of the model. That is, an MSE of 0.01 indicates the model can predict DNA methylation within 10% of its value typically. In this case, our dependent variables are always beta values for DNA methylation, therefore as explained in the paper, they are always on the scale of 0 to 1. By reporting the MSE, we specifically provide the readers with the error with which beta values of an independent cohort could be predicted in cross validations. We have explained this more clearly in the methods section of the revised manuscript and have included specific depictions of the data we are fitting.

We have now included pictorial examples in the text to explain what a specific MSE would look like. For example, Fig 1b is the figure shown below.

Figure 4a shown below is another example at the single promoter.

5. Lines 190-191. The authors claimed that "MAT2B and BHMT2 exhibit a larger contribution to local DNA methylation among the met cycle variables". However, in two cancer types, Colon cancer and Bladder cancer, the BHMT2 variable was selected 0% as shown in both Figures 2d and 2e.

The variable BHMT2 was removed from the models due to low overall expression in the cases of colon and bladder cancers, therefore it is missing from the plots for these two cancers. We thank the reviewer for bringing this to our attention. We have clarified this point in the figure legend. The statement in the lines mentioned by the reviewer refers to the average pattern across all cancers.

6. Lines 209-213, "Results from both Random Forest and Elastic Net algorithms confirmed a considerable contribution from the class "Methionine cycle enzymes" relative to other classes of variables (3rd strongest scoring among 8 classes, closely following

"Transcription Factors" and "Other SGO enzymes") according to both Elastic Net and Random Forest models (Figs. 3b,c)." I am not sure where does the rank comes from. But it seems to me in Figure 3C, Met-C is ranked 4th behind TFs, SGO and MUTs! I am confused.

The ranks were obtained by taking the average importance of each class across all 8 cancer types. In retrospect, we should have provided these average values in the manuscript as we agree with the reviewer that it is hard to tell the order from just looking at the circular plots. We have replaced the circular plots with box plots of the ranks of different functional classes in the revised manuscript to increase clarity of the comparisons.

7. Line 911. "All gene sets in Fig. 4 had Fisher's exact p-values <0.5". Is this a typo? If not, I would expect to see a very long list of pathway names in these figures.

Yes, this is in fact a typo. All gene sets had a Fisher's p-value <0.05. We appreciate the reviewer for bringing this to our attention and have corrected this in the revised version.

Minor:

What are the references at the end (lines 1040 and beyond) intended for?

We apologize for the confusion. The second set of references corresponds to citations in the online methods section of the manuscript. If the manuscript is accepted, we would re-format accordingly. Our intention was to make it easier for the reviewer.

Reviewer #2 (Remarks to the Author):

The idea of this paper is very interesting, and one must give a lot of credit to the authors for attempting to tease out potential mechanistic associations of what would be trans-interactions of genes regulating methylation, to the methylation changes in cancer. The problem(s) is that the paper is extremely confusing, which is partly because of the design, partly the writing, and partly the fact they are overlooking some key principles of cancer genetics as well as relevant literature. Criticisms are not in any particular order here:

We appreciate the reviewer's interest in our work and thank the reviewer for their insights. We have reworked the manuscript and its design substantially to improve the clarity. We also provide further clarifications below.

1. They would have to be extremely careful about arguing for mechanism. One cannot infer cause and effect from a study such as this. The very fact they are using machine learning means they are not defining a mechanistic link between one event and another, i.e. they don't know the causal flow here.

We agree with the reviewer that we can only suggest potential mechanistic links using our computational approach as we perform predictive modeling and not causal inference. In our case, we are hypothesizing that there is molecular and clinical information in the data that can explain tumor

DNA methylation. We do not intend to prove causation, however the algorithms utilized in our study are designed to accurately evaluate the predictive power of independent variables in explaining the dependent variable, therefore arguing for the suggestive direction of the associations is still valid from regression analysis based on biological understanding¹. We have further emphasized the predictive nature of our models in the results and discussion of the revised manuscript.

2. There is only one sample per tumor so they can't address intratumoral heterogeneity. At this point relating variability from person to person is fairly removed from real tumor heterogeneity, since this could be genetically driven.

We completely agree with the reviewer. In fact we stated this limitation inherent in our samples in lines 119-122 of the previous version of the manuscript (supplementary information in the revised version). We have also emphasized this point in the discussion section of the revised manuscript.

3. It is not an agnostic study in that they set out with a group of genes they think might be important. They don't compare to a similar "important group", like tyrosine kinases, or signaling receptors, etc.

We agree that this study is not completely agnostic. Unfortunately, multi-scale omics data sets contain too many variables with insufficient power for a completely unbiased study of this kind (e.g. if we considered every base pair in the genome, models of this nature do not have the power to identify driver mutations). We tried our best in this study to consider classes of biological variables that are known to affect DNA methylation and our main point was to quantify their relative contribution to being able to predict DNA methylation. We have mentioned this point in the revised text.

We further attempted to address this critical point in Supplementary Figure 7 by using randomized variables in our models and showed that the models are able to distinguish between noise and true associations. We thank the reviewer for this criticism and have included additional genes (variance-matched randomly selected genes) in the revised figure 1d shown below as controls.

4. 200 variables seems overspecified. Plus there are 18,000 models! How do you correct for this large number? The data in Figs 3 and 4 look highly overspecified.

We appreciate this concern. The algorithms we utilized have been developed to address the problem of overspecification in the presence of large number of variables. In fact, there is extensive documentation that both Elastic Net and Random Forest algorithms are able to perform well in presence of much larger variable sizes than what was used in our models (i.e. the cases of $p \gg n$)²⁻³. We have provided further clarification of this point in the revision. In fact, based on our cross validation at several points along the development of this study we are confident that the models did not suffer from over-fitting since if that were the case, then predictions on the test set samples would have been poor. This is emphasized in Fig 1 and several supplementary figures.

The 18000 models refer to prediction of DNA methylation at different locations across the genome. We apologize for the confusion but we are unsure about the specific concern the reviewer has regarding the 18,000 models or what the reviewer is referring regarding the data in Figs 3 and 4. We have however tried to provide further clarification in the text. Including several figures that attempt to address the problem of over specification. See for example the entirety of supplementary figure 8.

5. The methionine cycle itself is highly sensitive to metabolic changes in cancer, which we know are very much disturbed. So how do you know that the methylation changes and the methionine cycle aren't both disturbed by another extrinsic metabolic factor?

Our conclusions do not exclude this possibility and in fact we never suggest or intend to suggest that disruption of the methionine cycle is causal in tumorigenesis. We do however observe that changes in the methionine cycle are predictive of changes in DNA methylation which is the key conclusion of this study (and we believe a novel and important one). We have further clarified this point in the discussion section of the revised manuscript.

6. They do not make any tumor-normal comparisons.

The reviewer is correct. The study we've undertaken concerns variation in methylation and what determines it. Differential DNA methylation between tumor and corresponding normal tissue is however important to understand and has been extensively characterized⁴⁻⁶. We chose to analyze tumor tissues rather than normal tissues for a number of reasons: 1- DNA methylation is more variable in the context of tumors, and variation in tumor DNA methylation is biologically relevant since it can contribute to understanding cancer pathogenesis; 2- The TCGA data repository contains significantly more samples in all cases of cancerous tissues compared with corresponding normal, therefore it gives us higher statistical power to apply machine learning approaches with better success due to the large sample sizes.

7. They do not address batch effects and RNA quality in the TCGA data. These are potentially huge problems that could underlie their results.

We agree this is an important consideration. As described in the methods section, all RNA-seq data used in our study were obtained from the level-3 RNA-seqV2 and we only used the gene-normalized RSEM expression values. The RSEM algorithm takes RNA quality into account in the modeling step⁷

and this is standard practice in the hundreds of papers that have been published that analyze expression data from the TCGA.

We used the TCGA Batch Effects online tool (<http://bioinformatics.mdanderson.org/tcgabatcheffects>) to check for the existence of major batch effects in the data used in our study. For each cancer types in our study, both the DNA methylation and the RNA-seq batch effects were negligible (Dispersion Separability Criterion (DSC) score < 0.5). We thank the reviewer for pointing this out and have clarified this in the online methods section of the revised manuscript.

8. They do not address cell type heterogeneity, which is really surprising given the level of statistical expertise on the paper. In particular lymphocyte infiltration would likely cause big changes in methylation metabolism (they sue the Tet system, for example).

We agree with the reviewer that the cell types within tumors are heterogeneous and this is an important consideration and basic limitation of any analysis of bulk tumors. The goal of our computational models was to identify factors that can be potentially used to predict DNA methylation of overall tumor samples obtained in the clinic. We were focused on what variables could predict DNA methylation but agree that lymphocytes along with numerous other factors can cause changes in metabolism which are important and interesting but separate considerations. Rather, we focused on determining the impact of altered metabolism on DNA methylation levels and quantify its contribution to the overall variation observed in DNA methylation. We have discussed this limitation in the revised manuscript.

9. Could simply be degradation related, which also induces biochemical changes. What about cell type heterogeneity? This could have a major effect on everything they are talking about. Immune cells when activated are highly dynamic metabolically and also are known to use the Tet system.

We agree that these are important considerations and refer the reviewer to the response to comment #8.

10. They use bead chip only. That is very confusing since there are scattered CpG sites on these arrays. They also didn't use a block finder.

This is an important point. We considered Illumina bead chip methylation data due to larger sample size availability which enables this particular analysis. We mentioned this in the previous text and have further clarified this point in the discussion section of the revised text. We do acknowledge the limitation of these data as opposed to using whole genome bisulfite sequencing and we mentioned in the manuscript that the probe density across the genome is not uniform in these chips, thereby biasing our analysis toward genic regions of the epigenome (Supplementary figure 3). Although leading to under-representation of some potentially interesting regions of the genome, these chips do also serve to focus our analyses to better-characterized regions in the epigenome.

11. The 10 kb bin they use is highly biased toward what was curated in the first place, i.e. highly

functional regions of interest to 22 contributing labs, including DNase sensitivity, regulation etc., and is also enriched in CG dense regions.

The reviewer is correct. We acknowledge this limitation but we also assert that our analysis does not require that we know DNA methylation status down to a base-pair resolution for each tumor. Such data would allow for us to pursue even more analysis but the depth of the conclusions we could draw from the availability and subsequent analysis of such data we believe would be small compared to what we are able to achieve with these current data which are able to provide a number of novel conclusions about the determinants of DNA methylation.

12. As far as I can tell there is no replication set, e.g. working out the model in one group of tumors and checking in a different one. They just do out-of-bag calculations to quantify predictions.

We apologize for the confusion. The results were in fact cross-validated in independent testing sub-samples (3 fold cross validation was performed in each cancer type). This was in addition to the out-of-bag calculation on each training subset. We have revised the manuscript to emphasize the cross validation we have performed to make it easier for the readership to understand its scope.

We know a lot about huge methylation changes in glioma and leukemias due to mutations in specific methylation pathway genes. They don't start with that against known relevant data sets, and test that against papers that already address these issues, then go back and see if this same yardstick makes sense in the other tumor types.

We tried to account for genetic lesions including mutations and copy number alterations by including in the integrative models of each cancer type, all somatic mutations with a frequency of at least 5% and all CNVs with a frequency of at least 15% in the corresponding cancer types. We did indeed use findings from current understanding as positive controls in our models as described in page 16 of the manuscript (e.g. IDH1 mutation in glioma). In the case of leukemia, we were unable to study this cancer type unfortunately due to limited sample size. We thank the reviewer for pointing to this critical aspect of study design and have further clarified how mutations were accounted for in our study. We have also done our best to further show how these positive controls are used as confirmation in our study in the revision. See for example Fig. 4f,g and Supplementary Fig. 9 which now highlight the recovery of these known results. For example Fig. 4f and g, where the green arrows denote known results, are shown below.

13. They do not seem to be examining mutational status at all! One would then want to do a multivariate analysis against known mutations, and against other survival indicators.

We are very sorry for this misunderstanding. Our analysis does consider mutations as a source of variation in DNA methylation. We have revised the manuscript to illustrate and better emphasize this point. In the past, we have considered using multivariate Cox regression where appropriate to account for the impact of major mutations (and other factors) on overall survival. The issue here is that statistical power is reduced when accounting for subtypes in multivariate survival analysis. Nevertheless, our survival analysis is focused on the specific hypotheses we have made about tumors whose methylation can or cannot be predicted by the methionine cycle status and whether that status affects a clinical outcome.

To address the reviewer's concern, we repeated the survival analysis using multivariate cox regression for the cancer types where our univariate analyses had shown significant effects. In all cases tested, we still found a significant difference in survival ($p < 0.05$) between the two original groups of patients, even after accounting for relevant clinical factors and mutations. We have included the results of the multivariate survival analyses in the revised manuscript. The following is now added to the Online Methods section of the manuscript:

“ Multivariate cox regression

In the three cancer types (brain, liver, and kidney) where univariate analysis showed a highly significant difference in survival between the predictable and non-predictable groups as described above, and also the sample size allowed for sufficient power to perform multivariate analysis, we used relevant clinical and mutational factors as covariates and repeated the survival analysis. The following factors were individually tested as covariates in separate models of overall survival along with “predictability” status as the fixed effect: Brain cancer: all frequent somatic mutations (see Supplementary Table 1 for the complete list), histological diagnosis, age, gender, and initial weight; Liver cancer: all frequent somatic mutations (see Supplementary Table 1 for the complete list), tumor stage, history of other malignancies, and residual tumor; Kidney cancer: all frequent

somatic mutations (see Supplementary Table 1 for the complete list), age, and race. In each case, the results of regression using the “coxph()” function in R provided the p-value for the significance of the predictability status when modeling overall survival in the presence of covariates.”

References

1. Hastie, T.; Tibshirani, R.; Friedman, J., *The elements of statistical learning: Data Mining, Inference, and Prediction, Second Edition*. Springer: 2009.
2. Breiman, L., Random Forests. *Machine Learning* **2001**, *45* (1), 5-32.
3. Zou, H.; Hastie, T., Regularization and variable selection via the Elastic Net. *Journal of the Royal Statistical Society* **2005**, *Series B*, 301–320.
4. Gevaert, O.; Tibshirani, R.; Plevritis, S. K., Pancancer analysis of DNA methylation-driven genes using MethylMix. *Genome biology* **2015**, *16*, 17.
5. Timp, W.; Feinberg, A. P., Cancer as a dysregulated epigenome allowing cellular growth advantage at the expense of the host. *Nature reviews. Cancer* **2013**, *13* (7), 497-510.
6. Witte, T.; Plass, C.; Gerhauser, C., Pan-cancer patterns of DNA methylation. *Genome medicine* **2014**, *6* (8), 66.
7. Li, B.; Dewey, C. N., RSEM: accurate transcript quantification from RNA-Seq data with or without a reference genome. *BMC bioinformatics* **2011**, *12*, 323.

REVIEWERS' COMMENTS:

Reviewer #1 (Remarks to the Author):

The authors have addressed all my comments raised earlier. I am satisfied in general. but I am a bit confused with the newly added Figure 1 (d). the y-axis is rank. But the description claimed what being compared is the frequency of variable selection. i would assume lower rank means more importance, hence more frequent selection. Which seems contradict with the pattern shown in the figure.

Reviewer #2 (Remarks to the Author):

This is a revised paper, so I will confine my remarks to their response to the previous review:

1. Arguing for mechanism: I don't agree that simply setting up which are the dependent and independent variables establishes a causal flow. That's a classic epidemiological error, in fact. But what they can do is describe association without arguing causality and explaining that. See my remedy at the end.
2. OK
3. Non-agnostic nature of the design. Picking random genes doesn't satisfy this. They really need to pick some popular gene set as I mentioned in the first review. I think even if that also came out strongly associated, the results of the study would be important, but it would add caution to the interpretation.
4. OK
5. Similar to #1 actually, and see my remedy at the end.
- 6,7. OK
8. Tumor heterogeneity. See my remedy.
- 9-12. OK

On issue, #3, please check this.

On issues #1,5,8 (1 & 5 are the same really): add cautionary language to the abstract as that's what most people read.

With these changes I'd be really happy to see this come out. It's a great platform for discussion and new statistical analysis.

Reviewer #3 (Remarks to the Author):

Locasale and colleagues certainly provide a first-in-class approach attempting to comprehensively analyze the ultimate determinants of DNA methylation in human malignancies. The a priori unexpected large contribution of metabolism (i.e., metabolic genes involved in the methionine cycle that are constituents of one-carbon metabolism) in explaining epigenetic variation among individual tumors of the same cancer type definitely illustrates how metabolic remodeling can contribute to tumor epigenetic alterations.

It should be appreciated the great efforts that the authors have made in response to all the reviewers questions and concerns. The current revision certainly clarifies almost all the points the reviewers raised and should help the potential readers to understand the current manuscript.

Nonetheless, it should be noted that there is also some space for improvements for this paper; because the manuscript lacks a causal flow, the authors might at least elaborate on how (behavioral,

genetic and/or pharmacological) interventions on methionine metabolism will specifically impact (reverse?) those aberrant features of DNA methylation by “metabolically repairing” the disrupted regulation of one-carbon metabolic genes. In the same regard, the authors might discuss why and how regulation/dysregulation of the methionine cycle versus the folate cycle of the one-carbon metabolism might integrate the nutrient status to show specificity against tumor vs. normal tissue epigenetics.

Reviewer 1

The authors have addressed all my comments raised earlier. I am satisfied in general. but I am a bit confused with the newly added Figure 1 (d). the y-axis is rank. But the description claimed what being compared is the frequency of variable selection. i would assume lower rank means more importance, hence more frequent selection. Which seems contradict with the pattern shown in the figure.

We are delighted to hear that the previous round of revision are generally satisfactory to the review.

Regarding Figure 1d, we apologize for the confusion. The ranks are based on the frequency of variable selection in the case of Elastic Net, and the average variable importance in the case of Random Forest. The ranking as mentioned in the manuscript methods is in the decreasing order, with the most important group getting assigned the highest rank. We have further clarified this point in the text and figure legend.

Reviewer 2

This is a revised paper, so I will confine my remarks to their response to the previous review:

1. Arguing for mechanism: I don't agree that simply setting up which are the dependent and independent variables establishes a causal flow. That's a classic epidemiological error, in fact. But what they can do is describe association without arguing causality and explaining that. See my remedy at the end.

We agree with the reviewer and in the previous version, we acknowledged the predictive nature of our models and added emphasis that our results do not prove causality and one can only *suggest* direction based on prior biological understanding. To further address the reviewer's concern, we have now added additional cautionary notes on how the results should be interpreted in both the results and discussion sections.

2. OK

3. Non-agnostic nature of the design. Picking random genes doesn't satisfy this. They really need to pick some popular gene set as I mentioned in the first review. I think even if that also came out strongly associated, the results of the study would be important, but it would add caution to the interpretation.

We have attempted to further addressed this concern by repeating the DNA methylation modeling with addition of the following popular gene sets using the same approach as described in the random gene controls (Methods): Receptor tyrosine kinases (RTK), Receptor serine kinases (RSK), Toll like receptors (TLR), MAPK signaling pathway (MAPK), and WNT signaling pathway (WNT). With the exception of RTKs, each of these other popular gene sets score significantly lower

(Mann-Whitney p -value <0.0001) than our original gene expression variables. We have provided this result as an additional supplementary figure (Now Supplementary Fig. 3) and added cautionary language about the non-agnostic nature of our models and the possibility for some potentially interesting gene sets not being included (e.g. RTKs), perhaps explaining parts of the variation in DNA methylation that our models do not capture. Nonetheless, as stated in the manuscript, inference about relative contribution (i.e. a non-agnostic albeit reasonable approach in our minds) of the hundreds of variables that *are* included in our models is the main focus and goal behind our approach and we clearly state this when interpreting the results of our models throughout the manuscript.

Supplementary Figure 3. Comparison of our gene expression variables with popular gene families. Comparison of original gene expression variables with 5 popular gene sets was considered: Receptor tyrosine kinases (RTK), Receptor serine kinases (RSK), Toll like receptors (TLR), MAPK signaling (MAPK) and WNT signaling (WNT) families. The y-axis shows the average rank of each gene expression category based on average variable importance score across all Random Forest models of local DNA methylation in brain cancer (Error bars show the minimum and maximum value in each group). Significance of p -values associated with the Mann-Whitney test between the ranks across all models is shown (***: $p<0.0001$; see Methods). Results are shown for Random Forest models, but Elastic Net results showed the same pattern.

- 4. OK
- 5. Similar to #1 actually, and see my remedy at the end.

We have added additional cautionary language to the text.

- 6,7. OK
- 8. Tumor heterogeneity. See my remedy.

We have added additional cautionary language to the text.

- 9-12. OK

On issue, #3, please check this.

We thank the reviewer for noting specific suggestions and have done our best to revise the manuscript according to these instructions. See for example our response to concern #3.

On issues #1,5,8 (1 & 5 are the same really): add cautionary language to the abstract as that's what most people read.

We have added cautionary language to the manuscript and reworded the abstract as much as the limitations on the abstract word number allowed.

With these changes I'd be really happy to see this come out. It's a great platform for discussion and new statistical analysis.

We appreciate the time and effort that the reviewer put into helping us improve our work.

Reviewer 3

Locasale and colleagues certainly provide a first-in-class approach attempting to comprehensively analyze the ultimate determinants of DNA methylation in human malignancies. The a priori unexpected large contribution of metabolism (i.e., metabolic genes involved in the methionine cycle that are constituents of one-carbon metabolism) in explaining epigenetic variation among individual tumors of the same cancer type definitely illustrates how metabolic remodeling can contribute to tumor epigenetic alterations.

It should be appreciated the great efforts that the authors have made in response to all the reviewers questions and concerns. The current revision certainly clarifies almost all the points the reviewers raised and should help the potential readers to understand the current manuscript.

Nonetheless, it should be noted that there is also some space for improvements for this paper; because the manuscript lacks a causal flow, the authors might at least elaborate on how (behavioral, genetic and/or pharmacological) interventions on methionine metabolism will specifically impact (reverse?) those aberrant features of DNA methylation by “metabolically repairing” the disrupted regulation of one-carbon metabolic genes. In the same regard, the authors might discuss why and how regulation/dysregulation of the methionine cycle versus the folate cycle of the one-carbon metabolism might integrate the nutrient status to show specificity against tumor vs. normal tissue epigenetics.

We thank the reviewer for their positive assessment of our work and appreciate their insightful comments. Our results illustrate that higher levels of the methionine cycle enzymes and particularly MAT2B are associated with more “regulated”

methylation and higher survival. This suggests (but by no means confirms) potential for genetic or dietary intervention with methionine cycle components might enhance prognosis by altering the interaction between metabolism and epigenetics. We have added mention of this possibility in the discussion. Unfortunately it is a bit premature to speculate “metabolic repairing” beyond this suggestion, given that our models are inferential in nature and do not prove causal relationships. Furthermore, it remains to be experimentally tested whether a disordered DNA methylome can simply be fixed by such interventions since other factors might be playing critical roles as well. These studies are currently ongoing.